# MedSG-Bench: A Benchmark for Medical Image Sequences Grounding

**Jingkun Yue[1]   Siqi Zhang[1]   Zinan Jia[1]   Huihuan Xu[1]**
**Zongbo Han[1]   Xiaohong Liu[2]   Guangyu Wang[1]**[*]

[1]State Key Laboratory of Networking and Switching Technology, Beijing University of
Posts and Telecommunications   [2]South China Hospital, Medical School, Shenzhen University

## Abstract

Visual grounding is essential for precise perception and reasoning in multimodal large language models (MLLMs), especially in medical imaging domains. While existing medical visual grounding benchmarks primarily focus on single-image scenarios, real-world clinical applications often involve sequential images, where accurate lesion localization across different modalities and temporal tracking of disease progression (e.g., pre- vs. post-treatment comparison) require fine-grained cross-image semantic alignment and context-aware reasoning. To remedy the underrepresentation of image sequences in existing medical visual grounding benchmarks, we propose MedSG-Bench, the first benchmark tailored for **Med**ical Image **S**equences **G**rounding. It comprises eight VQA-style tasks, formulated into two paradigms of the grounding tasks, including 1) Image Difference Grounding, which focuses on detecting change regions across images, and 2) Image Consistency Grounding, which emphasizes detection of consistent or shared semantics across sequential images. MedSG-Bench covers 76 public datasets, 10 medical imaging modalities, and a wide spectrum of anatomical structures and diseases, totaling 9,630 question–answer pairs. We benchmark proprietary models (e.g., GPT-4o), general-purpose MLLMs (e.g., Qwen2.5-VL) and medical-domain specialized MLLMs (e.g., HuatuoGPT-vision), observing that even the advanced models exhibit substantial limitations in medical sequential grounding tasks. To advance this field, we construct MedSG-188K, a large-scale instruction-tuning dataset tailored for sequential visual grounding, and further develop MedSeq-Grounder, an MLLM designed to facilitate future research on fine-grained understanding across medical sequential images. We release all resources on `https://github.com/Yuejingkun/MedSG-Bench`

## 1   Introduction

Visual grounding is the key step that transforms MLLMs from coarse alignment between language expressions and corresponding visual regions to fine-grained visual understanding and reasoning[1]. For example, models like ChatGPT O3[2] often first identify image regions relevant to the questions during reasoning, which helps reduce hallucinations and enhances the trustworthiness of the results. This capability is particularly crucial in medical imaging, where understanding the semantic content of clinical text (e.g., radiology reports) and accurately localizing the corresponding pathological regions is essential for interpretable and reliable diagnosis[3, 4, 5, 6].

Currently, existing medical visual grounding benchmarks focus mainly on single-image scenarios [7, 8]. However, real-world clinical diagnosis inherently requires sequential image analysis. As

---

[*]Corresponding author: guangyu.wang24@gmail.com

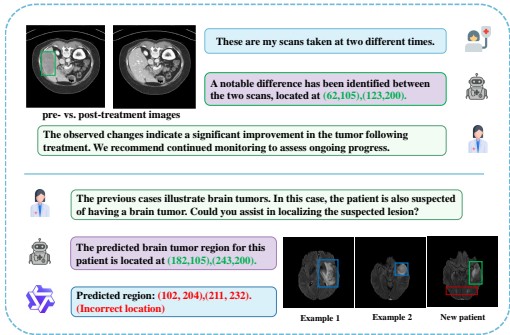

Figure 1: Examples of medical image sequences grounding.

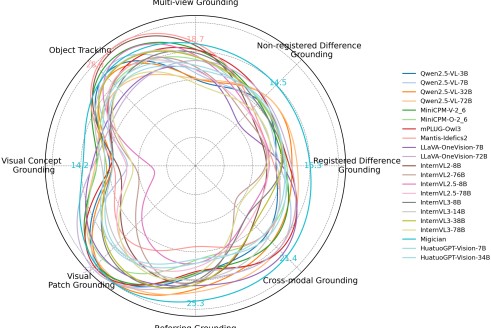

Figure 2: Comparing mainstream MLLMs on MedSG-Bench.

illustrated in Fig. 1, when assessing disease progression, clinicians routinely perform cross-image comparison (pre- vs. post-treatment images), tracking lesion evolution by analyzing changes in size, morphology, and signal intensity across longitudinal CT scans rather than relying solely on a single static image[9]. This essential practice of lesion localization and semantic alignment across multiple images forms the cornerstone of reliable clinical reasoning, yet remains underrepresented in current benchmarks.

To address this gap, we introduce MedSG-Bench, the first comprehensive benchmark specifically designed for medical visual grounding in sequential images. Built upon 76 publicly available medical imaging datasets, covering 10 imaging modalities, and 114 clinical tasks, our benchmark systematically evaluates cross-image grounding capability. Specifically, MedSG-Bench consists of eight carefully designed VQA-style tasks, organized into two grounding paradigms: 1) Image Difference Grounding, which targets the detection of differing regions between sequential images, and 2) Image Consistency Grounding, which focuses on discovering semantically consistent or shared regions across image sequences. This dual-paradigm grounding benchmark can evaluate the essential clinical competencies required for medical image analysis.

In summary, the contributions of this work are as follows:

1. We introduce MedSG-Bench, the first benchmark comprising 9,630 VQA-style samples specifically designed to evaluate the grounding capabilities of MLLMs in medical image sequences. The benchmark defines eight tasks grouped into two core paradigms, Image Difference Grounding and Image Consistency Grounding, which jointly serve to evaluate essential clinical competencies required for medical image analysis.

2. We conduct comprehensive evaluations of proprietary models (e.g., GPT-4o[10]), general-purpose MLLMs (e.g., Qwen2.5-VL[11]) and medical-domain specialized MLLMs (e.g., HuatuoGPT-Vision[12]) on MedSG-Bench. Our results (Fig. 2) show that all current MLLMs exhibit substantial limitations in fine-grained grounding of medical image sequences.

3. To promote progress in this underexplored area, we construct MedSG-188K, a large-scale instruction-tuning dataset tailored for grounding in medical image sequences. Based on this dataset, we further develop MedSeq-Grounder, and achieves state-of-the-art performance on MedSG-Bench.

## 2 Related work

### 2.1 Multimodal Large Language Models

Recent advances in multimodal large language models (MLLMs) have progressively extended their capabilities from coarse image-level understanding to fine-grained visual grounding[1, 13]. This progress has been primarily achieved through three main approaches: 1) instruction tuning with grounding supervision[14, 15], 2) integrating external localization modules[16, 17, 18, 19, 20, 21, 22] such as SAM[23] or Grounding DINO[24], and 3) leveraging vision tokenizers to enable perceive-then-understand paradigms[25, 26]. While these methods have significantly improved grounding

Table 1: Comparison between MedSG-Bench and other existing benchmarks in the medical field. FG denotes fine-grained annotation. * indicates the test set.

| Benchmark | Size | Task | Multi-modality | Multi-organ | Image-Sequence | FG | Max Length |
|---|---|---|---|---|---|---|---|
| **Understanding-oriented medical benchmarks** | | | | | | | |
| VQA-RAD[33] | 3K | 11 | ✓ | ✓ | ✗ | ✗ | 1 |
| SLAKE*[29] | 2K | 10 | ✓ | ✓ | ✗ | ✓ | 1 |
| OmniMedVQA[34] | 128K | 5 | ✓ | ✓ | ✗ | ✗ | 1 |
| GMAI-MMBench[30] | 26K | 18 | ✓ | ✓ | ✗ | ✓ | 1 |
| Medical-Diff-VQA*[31] | 70K | 7 | ✗ | ✗ | ✓ | ✗ | 2 |
| MMXU*[9] | 3K | 3 | ✗ | ✗ | ✓ | ✓ | 2 |
| **Grounding-oriented medical benchmarks** | | | | | | | |
| MS-CXR*[7] | 1K | 1 | ✗ | ✗ | ✗ | ✓ | 1 |
| MeCoVQA-G*[8] | 2K | 1 | ✓ | ✓ | ✗ | ✓ | 1 |
| MedSG-Bench | 9K | 8 | ✓ | ✓ | ✓ | ✓ | 6 |

accuracy within individual images, they largely overlook the clinically relevant and more complex setting of multi-image visual grounding. MC-Bench[27] first introduced the multi-context visual grounding task and Migician[28] is the first model to tackle this challenge in the natural image domain, enabling free-form and accurate grounding across multiple images. Building upon this paradigm, we extend the exploration to the medical domain, focusing on sequential visual grounding in clinically meaningful scenarios.

## 2.2 Medical MLLM Benchmarks

As shown in Table 1, benchmarks in the medical domain have progressed from early settings involving single-image and single-modality inputs to more advanced configurations covering multiple organs[29], cross-modal scenarios[30], and multi-image understanding[31, 9]. Some recent benchmarks[32] have also provided fine-grained annotations to enrich evaluation. However, these benchmarks primarily emphasize image-level understanding. Even when detailed annotations are available, they are typically utilized for classification or question answering tasks, rather than for explicit visual grounding. In contrast, grounding-oriented benchmarks remain scarce in the medical domain and are currently limited to single-image scenarios[7, 8]. To date, no medical benchmark has systematically explored sequential visual grounding, a capability that is essential for various clinical tasks such as cross-view lesion comparison, longitudinal disease progression tracking, and multi-phase imaging interpretation. To fill this gap, we propose MedSG-Bench, the first benchmark dedicated to fine-grained visual grounding in sequential medical images.

## 2.3 Temporal Medical Analysis

Recent studies have increasingly focused on incorporating temporal information to enhance the effectiveness of radiology retrieval and lesion progression detection. Some approaches[35, 36, 37] explicitly integrate temporal data as a feature within the model architecture, allowing the model to directly account for time-based changes in medical images. Other methods[38] treat temporal data as a dynamic semantic signal, improving the retrieval process by enabling the model to capture evolving patterns over time. Both strategies have shown promising results in downstream applications, particularly in medical report generation and disease progression analysis.

## 3 MedSG-Bench

In this section, we provide an in-depth overview of the careful design and development of MedSG-Bench, covering the rigorous collection and preprocessing of medical data, the systematic definition of tasks tailored for sequential visual grounding, and the presentation of detailed dataset statistics.

### 3.1 Data Collection and Preprocessing

#### 3.1.1 Dataset Review and Selection

As shown in Fig. 4, open data repositories, including Zenodo, Github, among others, were searched for medical image datasets. Data with permissive licenses (e.g., CC BY 4.0) that allow derivative

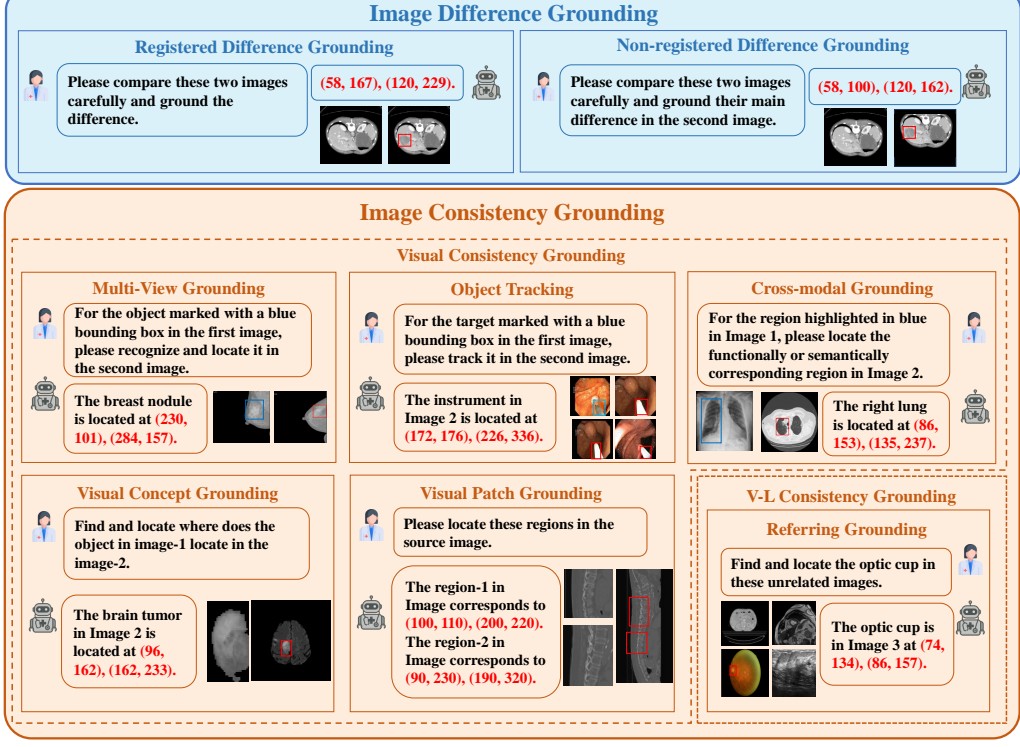

Figure 3: An illustration of medical image sequences grounding tasks included in MedSG-Bench.

works and redistribution were given priority during selection. We retained only those datasets that provided local annotations, such as segmentation masks or bounding boxes, which are essential for grounding-based tasks. To ensure mutual exclusivity among imaging cases, we cross-referenced dataset metadata and associated papers to identify and remove duplicated samples. Additionally, we performed a manual quality review to exclude images with poor visual clarity or unreliable annotations, and verified that all PHI (Protected Health Information) had been properly de-identified in the source datasets, thereby preserving the overall integrity and usability of the data.

### 3.1.2 Standardization

Medical imaging datasets exhibit high heterogeneity in format, resolution, intensity distribution, and metadata quality, with modality-specific characteristics that differ markedly from natural images. To mitigate this variability, we followed the preprocessing strategy proposed in [39], applying min-max normalization to rescale pixel intensities to a standardized range, thereby enabling more consistent downstream processing. To unify the data format, both 3D volumetric scans and video sequences were converted into 2D RGB images—achieved by slicing along anatomical axes or sampling frames at fixed intervals, respectively. All images were subsequently resized to 336×336 pixels, and each image was assigned a unique identifier encoding its imaging modality and associated task. Finally, all processed images were stored in lossless PNG format to preserve visual fidelity.

### 3.2 VQA tasks definition and generation

To facilitate fine-grained evaluation of visual grounding for sequential medical images, we define eight VQA-style tasks, organized into two complementary categories, including Image Difference Grounding and Image Consistency Grounding, which collectively capture both semantic changes and invariant features across image sequences, as illustrated in Fig. 3.

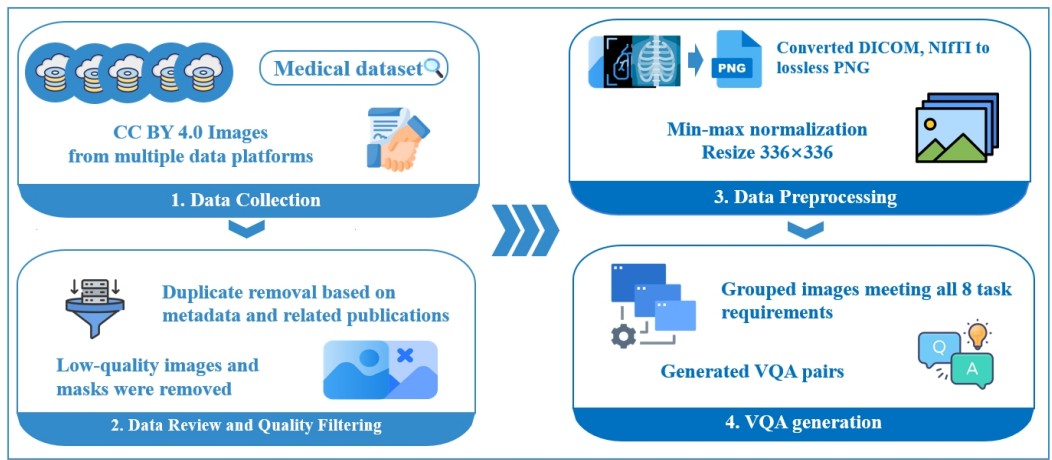

Figure 4: Overview of the MedSG-Bench construction protocol.

### 3.2.1 Image Difference Grounding

Image Difference Grounding focuses on detecting and localizing regions of changes across sequential images, enabling assessment of a model's ability to perceive subtle or clinically relevant variations.

**Task 1: Registered Difference Grounding**    Given a pair of spatially aligned (i.e., registered) images that are visually identical except for a single region, the model is designed to detect and localize the difference. To generate such image pairs in a controlled and scalable manner, we begin with a single medical image and introduce localized perturbations that simulate clinically meaningful variations, such as disease progression or treatment response. These perturbations comprise both geometric or appearance-based transformations (e.g., CutPaste[40]), and synthetic anomalies generated using state-of-the-art medical generative models[41, 42, 43]. To avoid the model learning shortcuts, such as associating a fixed image position with abnormalities, we randomize the ordering of image pairs, ensuring that either the normal or the abnormal image may appear in either position.

**Task 2: Non-registered Difference Grounding**    In clinical practice, medical images often exhibit spatial misalignments due to patient movement, scanner variability, or imperfect registration. This issue is particularly common when comparing medical images acquired from the same patient at different time points, where the lack of proper registration can lead to spatial shifts in organs or lesions, thereby potentially challenging models to distinguish real differences from registration artifacts. To better simulate such conditions and evaluate the model's robustness to Non-registered Difference Grounding, we extend Task 1 by introducing controlled spatial shifts: each image is randomly translated by up to 20 pixels along both the horizontal and vertical axes. The model is thus required to identify and accurately localize the primary difference between the two images while ignoring changes caused by misalignment.

### 3.2.2 Image Consistency Grounding

Image Consistency Grounding focuses on identifying and aligning invariant semantics across sequential medical images, which is essential for cross-view, cross-modal and cross-time alignment in clinical practice. Specifically, Image Consistency Grounding can be divided into two subcategories: 1) Visual Consistency Grounding (Task 3-7), which evaluates the model's ability to capture visual consistency across multiple images; 2) Vision-Language Consistency Grounding (Task 8), which involves aligning language-referenced information with multiple medical images.

**Task 3: Multi-View Grounding**    Medical images from different views often have geometric inconsistencies due to patient movement, scanning protocols, or anatomical deformation. To assess a model's ability to capture cross-view correspondence, we construct the Multi-View Grounding task using two implementation strategies. First, we repurpose existing multi-view datasets (e.g., VinDr-Mammo) by converting them into a VQA-style format. Second, we simulate multi-view

Table 2: Detailed statistics of MedSG-Bench.

| Task | #Datasets | #Modalities | #Clinical Tasks | Max Length |
|------|-----------|-------------|-----------------|------------|
| Registered Difference Grounding | 50 | 10 | 59 | 2 |
| Non-registered Difference Grounding | 50 | 10 | 58 | 2 |
| Multi-view Grounding | 30 | 4 | 75 | 3 |
| Object Tracking | 30 | 4 | 87 | 6 |
| Visual Concept Grounding | 49 | 10 | 87 | 2 |
| Visual Patch Grounding | 53 | 10 | 78 | 5 |
| Cross-modal Grounding | 24 | 4 | 28 | 4 |
| Referring Grounding | 9 | 8 | 28 | 3 |
| MedSG-Bench | 76 | 10 | 114 | 6 |

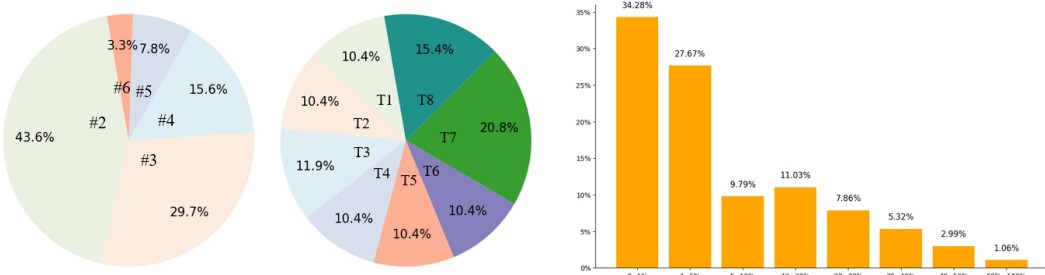

Figure 5: Proportions of image sequence length (**left**), data distribution across tasks (**middle**), and target-to-image size ratios (**right**) in MedSG-Bench.

scenarios by extracting three orthogonal slices (axial, sagittal, and coronal) from 3D medical volumes. Notably, the reference view is not fixed and may vary across different samples.

**Task 4: Object Tracking**  Accurately tracking anatomical structures or instruments across slices of medical images or frames of surgical video is essential in clinical workflows (e.g., lesion monitoring and intraoperative navigation). This task evaluates the model's ability to maintain consistent localization of a target object across sequential frames or slices. We construct this task using two types of data sources. First, we leverage existing surgical videos, where objects such as instruments or tissues are manually annotated across frames. Second, we simulate spatial tracking scenarios by slicing 3D medical volumes along a fixed anatomical axis, treating anatomical structures or lesions as trackable targets across ordered 2D slices.

**Task 5: Visual Concept Grounding**  In clinical scenarios, lesions can exhibit high variability in locations (e.g., across anatomical regions) and visual appearance due to imaging protocols or disease subtypes. This variability challenges models to learn robust target representations based on pathological features, rather than over-relying on spatial biases. This task evaluates the model's ability to recognize and localize a visually distinct and semantically coherent concept, including both pathological findings such as tumors and anatomical structures such as organs or tissue subtypes, within a complex medical image. The model is provided with a reference image in which the concept appears under idealized conditions, and must identify the corresponding instance in a target image with greater visual clutter and contextual complexity. To construct this task, the reference concept is extracted from the target using segmentation masks to ensure semantic consistency.

**Task 6: Visual Patch Grounding**  Precisely distinguishing nearly identical anatomical structures (e.g., separating tumor margins from adjacent vasculature) is essential for image-guided interventions and radiotherapy planning, where subtle visual distinctions determine procedural success. Therefore, we design this task evaluates the model's ability to match a local image patch to its original location within a larger image. It poses significant challenges in contexts where structures like vertebral segments (e.g., T1 to T12) exhibit nearly identical appearances. To construct this task, we initially sample 15 patches per image and manually select up to five based on foreground richness, including organ boundaries, lesion areas, or diagnostically relevant fine structures. The rest are discarded. This

selective sampling ensures that each retained patch presents a non-trivial grounding challenge while avoiding visually homogeneous regions.

**Task 7: Cross-modal Grounding**    In clinical practice, the same patient is often examined using different imaging modalities such as CT, X-ray, or MRI, each highlighting distinct but complementary aspects of anatomical structures or pathologies. This task assesses the model's ability to ground semantically or functionally equivalent regions across differing imaging contexts. Given a reference region from one image, the model is required to identify the corresponding region in a target image that may differ in imaging modality (e.g., CT versus MRI) or contrast type (e.g., T1-weighted versus T2-weighted MRI). Region pairs are manually curated based on metadata such as modality type and annotated labels to ensure semantic alignment and multimodal consistency.

**Task 8: Referring Grounding**    Clinicians often describe findings or refer to specific regions using natural language expressions. Enabling models to accurately interpret and associate such expressions with visual content is essential for enhancing interpretability, supporting human-AI collaboration, and building reliable decision support systems. Considering the prevalence of partially labeled data in medical imaging, we carefully curate candidate image sets to ensure that the images are semantically unrelated. This reduces the risk of referential ambiguity caused by overlapping content or latent correlations among images.

## 3.3   Data description

We curated a total of 76 publicly available datasets under permissive licenses, prioritizing those released with open CC-BY terms to ensure broad accessibility. As summarized in Table 2, MedSG-Bench spans 10 medical imaging modalities (CBCT, CT, CTA, Colonoscopy, Dermoscopy, Endoscopy, Fundus, MRI, US, X-ray) and and encompasses 114 distinct clinical tasks, covering a wide range of anatomical regions and disease types. The benchmark contains 9,630 visual question answering pairs, derived from 24,341 medical images, designed to assess fine-grained grounding capabilities across diverse clinical contexts. In addition to task coverage, we also provide detailed statistics on the proportion of image sequence lengths, data distribution, and target-to-image size ratios (lesions or anatomical abnormalities are often subtle, localized, and small in size), offering a comprehensive overview of the benchmark's complexity and representativeness in Fig. 5.

# 4   MedSG-188K and MedSeq-Grounder

## 4.1   MedSG-188K

The construction of MedSG-188K is based on the eight tasks defined by MedSG-Bench. To ensure diversity in VQA-style queries, we first crafted seed instruction templates tailored to the specific characteristics of each task, capturing the nuanced demands of distinct clinical scenarios. To mitigate potential bias and enhance linguistic diversity, we employed multiple large language models (LLMs), including GPT-4[44], Claude[45], and DeepSeek[46], to expand the seed instruction templates. These models collectively generated ten diverse free-form instruction variants per task by systematically varying the phrasing, contextual framing, and query structure. For each medical image sequence, one of the instruction templates was randomly selected and populated with task-specific content to generate diverse question-answer pairs. Using this pipeline, we constructed a total of 188,163 VQA-style samples, derived from 324,359 medical images. The distribution of sequence lengths, data volume is summarized in Fig. 6.

## 4.2   MedSeq-Grounder

MedSeq-Grounder is developed based on the Qwen2.5-VL-7B model[11] and trained using the LLaMA-Factory framework[47]. The training is performed with a global batch size of 64 over 15,000 steps, using a learning rate of 5e-6 and 4×A40-48G GPUs.

Table 3: Performance of different MLLMs on MedSG-Bench. IDG: Image Difference Grounding; ICG: Image Consistency Grounding; RDG: Registered Difference Grounding; NRDG: Non-registered Difference Grounding; MV: Multi-view Grounding; OT: Object Tracking; VCG: Visual Concept Grounding; VPG: Visual Patch Grounding; CMG: Cross-modal Grounding; RG: Referring Grounding; Avg.: Average; IoU and acc@0.5 for all results are shown, all numbers are in percentages.

| Model | Size | IDG | | ICG | | | | | | Avg. |
|---|---|---|---|---|---|---|---|---|---|---|
| | | RDG | NRDG | MV | OT | VCG | VPG | CMG | RG | |
| **Proprietary MLLMs** | | | | | | | | | | |
| GPT-4o[10] | – | 2.42 | 3.45 | 16.51 | 28.19 | 13.18 | 38.05 | 16.02 | 23.08 | 17.70 |
| | | 0.40 | 0.20 | 8.62 | 23.90 | 4.70 | 26.40 | 4.95 | 18.02 | 10.60 |
| Claude Sonnet 4[45] | – | 0.67 | 0.81 | 12.56 | 23.11 | 6.93 | 27.44 | 9.04 | 19.57 | 12.51 |
| | | 0.00 | 0.10 | 3.57 | 16.50 | 1.40 | 13.80 | 1.80 | 10.80 | 5.76 |
| Gemini 2.5 Pro[48] | – | 9.36 | 7.29 | 14.26 | 19.32 | 14.94 | 41.11 | 24.44 | 28.12 | 20.66 |
| | | 3.20 | 2.00 | 6.71 | 13.80 | 10.70 | 49.20 | 28.12 | 22.67 | 15.61 |
| **General-purpose MLLMs** | | | | | | | | | | |
| Qwen2.5-VL[11] | 3B | 0.59 | 1.62 | 7.12 | 21.32 | 6.98 | 27.36 | 10.02 | 12.99 | 10.94 |
| | | 0.30 | 1.30 | 3.90 | 16.80 | 0.80 | 3.40 | 1.65 | 6.82 | 4.20 |
| Qwen2.5-VL[11] | 7B | 0.88 | 1.25 | 8.48 | 22.41 | 4.22 | 28.87 | 16.29 | 12.58 | 12.31 |
| | | 0.30 | 0.00 | 3.73 | 17.80 | 1.00 | 5.70 | 4.45 | 6.21 | 4.90 |
| Qwen2.5-VL[11] | 32B | 2.69 | 3.48 | 7.35 | 19.12 | 6.53 | 26.92 | 12.59 | 18.71 | 12.47 |
| | | 1.40 | 1.20 | 2.61 | 13.40 | 1.30 | 7.10 | 4.90 | 11.67 | 5.71 |
| Qwen2.5-VL[11] | 72B | 4.37 | 3.46 | 7.22 | 13.11 | 10.33 | 26.45 | 16.32 | 20.19 | 13.35 |
| | | 2.60 | 0.80 | 2.78 | 7.70 | 3.50 | 6.30 | 7.00 | 14.10 | 6.12 |
| MiniCPM-V-2_6[49] | 8B | 1.36 | 1.50 | 15.82 | 24.03 | 9.90 | 28.65 | 12.72 | 12.44 | 13.24 |
| | | 0.00 | 0.00 | 5.20 | 18.50 | 2.10 | 12.20 | 3.30 | 3.64 | 5.27 |
| MiniCPM-O-2_6[50] | 8B | 1.69 | 1.63 | 12.11 | 15.25 | 9.88 | 22.96 | 9.53 | 8.82 | 10.12 |
| | | 0.10 | 0.00 | 2.43 | 9.60 | 1.70 | 9.20 | 2.35 | 2.02 | 3.23 |
| mPLUG-Owl3[51] | 7B | 2.12 | 2.55 | 15.64 | 15.62 | 6.80 | 30.42 | 17.06 | 11.92 | 13.22 |
| | | 0.00 | 0.00 | 3.64 | 4.40 | 0.80 | 3.60 | 4.80 | 5.47 | 3.19 |
| Mantis-Idefics2[52] | 8B | 0.49 | 0.62 | 18.69 | 28.04 | 6.27 | 10.26 | 9.59 | 6.05 | 9.90 |
| | | 0.00 | 0.00 | 8.59 | 23.50 | 0.50 | 1.10 | 0.95 | 0.54 | 3.91 |
| LLaVA-OneVision[53] | 7B | 1.09 | 0.01 | 9.26 | 10.50 | 11.33 | 22.20 | 19.08 | 17.11 | 12.39 |
| | | 0.00 | 0.00 | 1.13 | 3.20 | 1.80 | 5.30 | 6.70 | 5.67 | 3.47 |
| LLaVA-OneVision[53] | 72B | 2.58 | 2.87 | 11.74 | 9.61 | 10.95 | 32.38 | 16.24 | 15.43 | 13.21 |
| | | 0.80 | 0.90 | 1.39 | 2.30 | 3.30 | 20.30 | 5.40 | 6.68 | 5.18 |
| InternVL3[54] | 8B | 1.07 | 1.20 | 14.36 | 13.30 | 6.43 | 18.73 | 4.73 | 15.16 | 9.26 |
| | | 0.30 | 0.00 | 4.42 | 6.50 | 0.90 | 4.60 | 1.15 | 7.42 | 3.19 |
| InternVL3[54] | 14B | 0.66 | 0.71 | 13.24 | 19.77 | 8.60 | 13.17 | 10.87 | 14.57 | 10.53 |
| | | 0.00 | 0.00 | 5.31 | 13.00 | 2.10 | 2.40 | 3.70 | 7.76 | 4.41 |
| InternVL3[54] | 38B | 0.98 | 1.76 | 12.99 | 19.27 | 7.63 | 17.76 | 6.47 | 16.59 | 10.37 |
| | | 0.10 | 0.20 | 4.79 | 13.60 | 2.10 | 2.90 | 1.75 | 10.05 | 4.44 |
| InternVL3[54] | 78B | 0.20 | 0.53 | 6.35 | 13.03 | 3.57 | 11.81 | 3.34 | 12.76 | 6.44 |
| | | 0.00 | 0.00 | 2.43 | 8.00 | 0.90 | 2.50 | 0.85 | 8.10 | 2.90 |
| Migician[28] | 7B | 15.26 | 14.49 | 18.16 | 21.38 | 14.23 | 28.87 | 21.41 | 25.30 | 20.29 |
| | | 7.80 | 6.10 | 7.84 | 14.90 | 7.20 | 13.70 | 12.15 | 18.02 | 11.39 |
| **Medical-domain specialized MLLMs** | | | | | | | | | | |
| MedGemma[55] | 4B | 0.45 | 0.84 | 7.80 | 26.82 | 11.31 | 26.59 | 5.92 | 10.01 | 10.55 |
| | | 0.00 | 0.00 | 4.53 | 22.40 | 0.90 | 15.40 | 0.50 | 1.01 | 4.82 |
| HuatuoGPT-Vision[12] | 7B | 1.35 | 1.84 | 10.42 | 14.57 | 7.99 | 15.52 | 9.46 | 9.60 | 8.97 |
| | | 0.00 | 0.20 | 2.78 | 9.20 | 0.80 | 2.30 | 2.15 | 1.82 | 2.36 |
| HuatuoGPT-Vision[12] | 34B | 1.44 | 2.15 | 9.41 | 13.25 | 6.43 | 14.53 | 10.60 | 8.60 | 8.57 |
| | | 0.00 | 0.00 | 1.65 | 8.30 | 0.70 | 1.40 | 2.60 | 1.75 | 2.09 |
| MedSeq-Grounder (Ours) | 7B | **83.29** | **83.72** | **55.03** | **62.10** | **74.11** | **85.25** | **78.77** | **60.43** | **72.55** |
| | | 93.20 | 94.10 | 60.19 | 67.20 | 82.60 | 98.80 | 82.75 | 65.59 | 79.71 |

# 5 Experiments

## 5.1 Experiment setup

In this study, we evaluate model performance under a zero-shot setting, where the models were prompted to perform inference without access to in-context examples. We use average Intersection over Union (IoU) and ACC@0.5 as the evaluation metric.

## 5.2 Models

We benchmark a diverse collection of state-of-the-art MLLMs on MedSG-Bench, including 1) proprietary models, 2) general-purpose models that have extended capabilities in the medical domain, and 3) medical-domain specialized models that are meticulously trained for clinical medicine. All models support image sequence input and span parameter scales from approximately 3 billion to 70 billion. For public models, we use publicly released checkpoints from their official Hugging Face repositories[56], selecting the latest or best-performing version within each model family. For proprietary models, we utilize their respective APIs to access the latest available versions.

**Proprietary MLLMs**   We evaluate GPT-4o[10], Claude Sonnet 4[45], and Gemini 2.5 Pro[48].

**General-Purpose MLLMs**   We evaluate Qwen2.5-VL (3B, 7B, 32B, 72B)[11], MiniCPM-V-2_6[49], MiniCPM-O-2_6[50], mPlug-owl3[51], Mantis-Idefics2[52], llava_onevision (7B, 72B)[53], internvl2 (8B, 78B)[57, 58], internvl2_5 (8B, 78B)[59], internvl3 (8B, 14B, 38B, 78B)[54]. For grounding-oriented MLLMs, we evaluate Migician[28], which supports free-form multi-image grounding and has strong instruction-following capability.

**Medical-domain specialized MLLMs**   We evaluate HuatuoGPT-Vision (7B, 34B)[12], which is built on a large-scale and high-quality medical VQA dataset, PubMedVision, as well as other models such as MedGemma (4B)[55], LLaVA Med v1.5 (7B)[60], and BiMediX2 (8B)[61], which are also trained on specialized medical datasets for medical-domain tasks.

In our evaluation process, we carefully considered the potential impact of inconsistencies in coordinate formats across different models. To address this, we consulted the official documentation or papers for each baseline model to determine the expected coordinate format. For instance, the InternVL series models normalize coordinates to the range [0, 1000], while Qwen2.5-VL supports absolute coordinate output, and Gemini 2.5 Pro uses the format [y_min, x_min, y_max, x_max].

## 5.3 Main Results

Based on the evaluation results presented in Table 3 and  5, we have some findings as follows:

**Grounding in medical image sequences is still challenging for all MLLMs**   Our MedSG-Bench provides a comprehensive multitask challenge, revealing that even the top-performing model Gemini-2.5-pro is limited to the average IoU of 20.66% and Acc@0.5 of 15.61% in zero-shot setting. In particular, most MLLMs struggle with the Image Difference Grounding task. Moreover, the most advanced models do not consistently excel across all tasks, for example, while Gemini-2.5-pro achieves relatively high accuracy on the cross-modal grounding task, its performance on multi-view grounding or object tracking remains notably lower than Mantis and GPT-4o, highlighting the challenge of generalization across diverse grounding scenarios. With instruction tuning on our MedSG-188K dataset, the proposed MedSeq-Grounder achieves state-of-the-art performance across all tasks, demonstrating its effectiveness and robustness in sequential medical visual grounding.

**All MLLMs exhibit limitations in detecting small medical targets**   Small target recognition is a critical challenge in the medical domain, we further categorized the targets into three groups based on their bounding box area ratio: small (0-1%), medium (1-10%), and large (>10%). Table  7 demonstrates that most MLLMs exhibit substantially reduced performance on small targets, underscoring their limitations in precise medical sequential grounding. In contrast, MedSeq-Grounder consistently achieves strong performance across all target sizes, demonstrating its robustness grounding capability in clinically challenging scenarios.

**Medical-domain specialized models are often worse than general-purpose models** While specialist models are explicitly developed for the medical domain, they often underperform non-specialist open-source models. For example, HuatuoGPT-Vision-7B, lags behind Qwen2.5-VL-7B by 3.34% in average IoU and 2.54% in Acc@0.5 on MedSG-Bench. Notably, it even performs worse than the smaller-sized Qwen2.5-VL-3B model. This performance gap may be attributed to the nature of training data used for domain adaptation. Most existing medical instruction-tuning datasets focus predominantly on image-level understanding tasks, such as classification or report summarization. While HuatuoGPT-Vision is built upon Qwen-VL, its further tuning on understanding-centric medical data appears to have degraded its grounding capability. This reflects a case of catastrophic forgetting, where the model's original ability for spatial alignment is compromised due to continued learning on tasks that lack grounding supervision.

**Larger or newer models do not guarantee improved grounding performance** Although model scale and recency are commonly associated with improved performance, we find that larger or more recently released models do not necessarily exhibit stronger grounding capabilities in medical image sequences. For instance, InternVL2.5-8B and InternVL3-8B both underperform compared to the earlier InternVL2-8B model, despite architectural updates and increased pretraining. Similarly, MiniCPM-O-2_6 lags behind MiniCPM-V-2_6, highlighting that newer instruction-tuned variants may sacrifice grounding performance in favor of improvements on general-purpose understanding tasks. In some cases, such as with the InternVL family, even the 70B-scale model yields worse results on MedSG-Bench compared to its 8B counterpart, indicating that grounding ability may not scale proportionally with model size. These results suggest that many recent models are primarily optimized for high-level semantic tasks, such as open-ended QA or captioning, and are trained on instruction-tuning datasets that provide little to no supervision for spatial localization or visual grounding. This observation further underscores the importance of dedicated benchmarks like MedSG-Bench, which are specifically designed to evaluate fine-grained grounding and spatial alignment across sequential medical images.

## 6 Conclusion

This work introduces MedSG-Bench, the first benchmark specifically designed to evaluate the fine-grained visual grounding capabilities of MLLMs in sequential medical images. Through systematic evaluations on eight clinically inspired grounding tasks, we find that all current MLLMs exhibit substantial limitations in medical image sequences grounding. To address these challenges, we construct a grounding instruction-tuning dataset, MedSG-188K, and develop MedSeq-Grounder. We hope our benchmark, dataset, and model will together advance the development of visual grounding in medical image sequences.

## 7 Limitations and Future Work

While MedSG-Bench is constructed from a wide range of publicly available datasets, it does not include private real-world clinical data such as longitudinal studies, multi-timepoint diagnostics, or follow-up imaging records. This limits its ability to fully capture the temporal complexity and diagnostic continuity inherent in actual clinical workflows. Meanwhile, MedSeq-Grounder is a task-specific model for medical image sequences grounding. Directly fine-tuning may reduce its performance on other tasks such as free-text QA.

In future work, we plan to collaborate with medical institutions to incorporate authentic clinical data, including patient trajectories across multiple visits and imaging sessions, to enhance the benchmark's realism and clinical applicability. And we plan to continue training the model on broader medical instruction data beyond grounding tasks to enhance its general multimodal capabilities.

## Acknowledgements

This study was funded by the National Natural Science Foundation of China (grants T2522008, 62272055, 82522048, 62425112 and 624B2100), New Cornerstone Science Foundation through the XPLORER PRIZE, Xiaomi Foundation.

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

# A  Dataset Details

In this section, we provide the detailed datasets used in MedSG-Bench, including the name of the dataset, the modality, the dimension of data, and the accessible links. As shown in Table 4, MedSG-Bench is constructed from 76 datasets across 10 medical image modalities.

Table 4: Detailed datasets information in MedSG-Bench.

| Dataset | Modality | Dim | Accessible links |
|---|---|---|---|
| 4C2021[62] | CT | 3D | https://aistudio.baidu.com/datasetdetail/89548 |
| AbdomenCT1K[63] | CT | 3D | https://github.com/JunMa11/AbdomenCT-1K |
| ACDC[64] | MRI | 3D | https://humanheart-project.creatis.insa-lyon.fr/database/ |
| AMOS22[65] | CT, MRI | 3D | https://amos22.grand-challenge.org/ |
| ATM22[66] | CT | 3D | https://atm22.grand-challenge.org/ |
| Atria Segmentation[67] | MRI | 3D | https://www.cardiacatlas.org/atriaseg2018-challenge/atria-seg-data/ |
| AutoLaparo[68] | Colonoscopy | 2D | https://autolaparo.github.io/ |
| BAGLS[69] | Endoscopy | 2D | https://www.kaggle.com/datasets/gomezp/benchmark-for-automatic-glottis-segmentation |
| BraimMRI[70] | MRI | 3D | https://www.kaggle.com/datasets/masoudnickparvar/brain-tumor-mri-dataset |
| BrainPTM[71][72] | MRI | 3D | https://brainptm-2021.grand-challenge.org/ |
| BraTS2020[73][74][75] | MRI | 3D | https://service.tib.eu/ldmservice/dataset/brats2020 |
| BUSI[76] | US | 2D | https://scholar.cu.edu.eg/?q=afahmy/pages/dataset |
| CAD-PE[77] | CT | 3D | https://ieee-dataport.org/open-access/cad-pe |
| CAMUS[78] | US | 2D | https://www.creatis.insa-lyon.fr/Challenge/camus/ |
| Cause07[79] | MRI | 3D | https://cause07.grand-challenge.org/ |
| CBCT3D[80][81] | CBCT | 3D | https://toothfairy.grand-challenge.org/ |
| Chestimage[82] | X-Ray | 2D | https://tianchi.aliyun.com/dataset/83075 |
| CMRxMotions[83] | MRI | 3D | https://www.synapse.org/Synapse:syn28503327/ |
| COVID-19[84] | CT | 3D | https://medicalsegmentation.com/covid19/ |
| COVID19CTscans[85] | CT | 3D | https://zenodo.org/records/3757476 |
| COVID-19-20[86] | CT | 3D | https://covid-segmentation.grand-challenge.org/ |
| Covid19cxr[87] | X-ray | 2D | https://github.com/ieee8023/covid-chestxray-dataset |
| Cranium[88] | CT | 3D | https://tianchi.aliyun.com/dataset/82967 |
| CT-ORG[89] | CT | 3D | https://www.cancerimagingarchive.net/collection/ct-org/ |
| CTSpine1K[90] | CT | 3D | https://github.com/MIRACLE-Center/CTSpine1K |
| CVC-ClinicDB[91] | Colonoscopy | 2D | https://polyp.grand-challenge.org/CVCClinicDB/ |
| DRISHTI-GS[92] | Fundus | 2D | https://www.kaggle.com/datasets/lokeshsaipureddi/drishtigs-retina-dataset-for-onh-segmentation |
| EMIDEC[93] | MRI | 3D | https://emidec.com/dataset |
| EndoTect2020[94] | Colonoscopy | 2D | https://osf.io/mh9sj/ |
| EndoVis15[95] | Colonoscopy | 2D | https://endovis.grand-challenge.org/ |
| EndoVis2017[96] | Colonoscopy | 2D | https://endovissub2017-roboticinstrumentsegmentation.grand-challenge.org/ |
| GAMMA[97][98][99] | Fundus | 2D | https://gamma.grand-challenge.org/Home/ |
| HaN-Seg[100] | CT, MRI | 3D | https://zenodo.org/records/7442914 |
| Hvsmr2016[101] | MRI | 3D | http://segchd.csail.mit.edu/data.html |
| I2CVB[102] | MRI | 3D | https://i2cvb.github.io/ |
| InSTANCE2022[103][104] | CT | 3D | https://instance.grand-challenge.org/ |
| iseg2017[105] | MRI | 3D | https://iseg2017.web.unc.edu/download/ |
| ISIC2018[106][107] | Dermoscopy | 2D | https://challenge.isic-archive.com/data/#2018 |
| ISLES-ATLAS[108] | MRI | 3D | https://atlas.grand-challenge.org/ |
| ISLES-MM[108] | MRI | 3D | https://isles22.grand-challenge.org/ |
| JSRT[109] | X-ray | 2D | http://imgcom.jsrt.or.jp/minijsrtdb/ |
| KvasirInstrument[110] | Colonoscopy | 2D | https://datasets.simula.no/kvasir-instrument/ |

| | | | |
|---|---|---|---|
| LMSLS[111] | MRI | 3D | https://smart-stats-tools.org/lesion-challenge-2015 |
| LUNA16[112] | CT | 3D | https://luna16.grand-challenge.org/Download/ |
| MMWHS[113][114][115][116] | CT, MRI | 3D | https://www.ub.edu/mnms/ |
| MRSpineSeg[117][118] | MRI | 3D | https://mosmed.ai/datasets/covid19_1110 |
| MSD02[119] | MRI | 3D | http://medicaldecathlon.com/ |
| MSD04[120] | MRI | 3D | http://medicaldecathlon.com/ |
| MSD05[120] | MRI | 3D | http://medicaldecathlon.com/ |
| MyoPS2020[113][114][116] | MRI | 3D | https://zmiclab.github.io/zxh/0/myops20/ |
| NCI-ISBI2013[121] | MRI | 3D | https://www.cancerimagingarchive.net/analysis-result/isbi-mr-prostate-2013/ |
| PadChest[122] | X-ray | 2D | https://bimcv.cipf.es/bimcv-projects/padchest/ |
| PALM[123] | Fundus | 2D | https://ieee-dataport.org/documents/palm-pathologic-myopia-challenge |
| Parse2022[124] | CT | 3D | https://parse2022.grand-challenge.org/Dataset/ |
| PCXA[125][126] | X-ray | 2D | https://lhncbc.nlm.nih.gov/LHC-downloads/downloads.html |
| PDDCA[127] | CT | 3D | https://www.imagenglab.com/newsite/pddca/ |
| Pelvic1K[128] | CT | 3D | https://zenodo.org/record/4588403 |
| Promise09[129] | MRI | 3D | https://www.na-mic.org/wiki/Training_Data_Prostate_Segmentation_Challenge_MICCAI09 |
| PROMISE12[130] | MRI | 3D | https://zenodo.org/records/8026660 |
| QaTa-COV19[131][132][133][134][135] | X-ray | 2D | https://www.kaggle.com/datasets/aysendegerli/qatacov19-dataset |
| QUBIQ2020[136] | CT | 2D | https://qubiq.grand-challenge.org/ |
| REFUGE[99][137] | Fundus | 2D | https://refuge.grand-challenge.org/ |
| RIGA+[138] | Fundus | 2D | https://zenodo.org/records/6325549 |
| RIM_ONE[139] | Fundus | 2D | https://github.com/miag-ull/rim-one-dl |
| SegRap2023[140] | CT | 2D | https://segrap2023.grand-challenge.org/dataset/ |
| SegTHOR[141] | CT | 3D | https://competitions.codalab.org/competitions/21145 |
| SIIM-ACR[142] | X-ray | 2D | https://www.kaggle.com/c/siim-acr-pneumothorax-segmentation |
| SKI10[143] | MRI | 3D | https://ski10.grand-challenge.org/ |
| SLAWT[144] | MRI | 3D | http://stacom.cardiacatlas.org/ |
| TBAD[145] | CTA | 3D | https://www.kaggle.com/datasets/xiaoweixumedicalai/imagetbad |
| TN-SCUI[146] | US | 2D | https://tn-scui2020.grand-challenge.org/ |
| VESSEL12[147] | CT | 3D | https://vessel12.grand-challenge.org/ |
| VINDR-Mammo[148] | X-ray | 2D | https://www.physionet.org/content/vindr-mammo/1.0.0/ |
| Verse19[149][150] | CT | 3D | https://github.com/anjany/verse |
| WMH[151] | MRI | 3D | https://dataverse.nl/dataset.xhtml?persistentId=doi:10.34894/AECRSD |
| WORD[152] | CT | 3D | https://github.com/HiLab-git/WORD |

# B  Template for Instruction Data Generation

---

**Template for Instruction Data Generation**

**Task1**
"Please examine these two images and provide the coordinates of the area where they differ."
"Compare both images closely and share the coordinates of the discrepancy."
"Look at these two images and tell me the coordinates of the difference between them."
"Carefully analyze these images and provide the coordinates of their difference."
"Examine the two images and give me the coordinates of the region where they differ."
"Can you find the differences between these two images and give me the coordinates?"
"Please inspect these two images and indicate the coordinates of their difference."
"Compare the two images and identify the coordinates of the difference."
"Look closely at the two images and provide the coordinates where they differ."
"Analyze both images and provide the coordinates of the difference between them."
**Task2**
"Compare these two images carefully and give me the coordinates of their real difference in the second image. Find it and locate it in the second image."
"Please examine both images and identify the real difference that appears in the second one. Provide the coordinates of that difference."
"Carefully analyze the two images. What is the actual visual change in the second image? Mark its coordinates precisely."
"Spot the true difference in the second image when compared with the first. Return the bounding box of that change."
"Look at the two images side by side. What is the meaningful change introduced in the second image? Output its location."
"Your task is to detect the actual difference in the second image compared to the first and report its position in coordinates."
"Inspect the two images and tell me where the real change is in the second one. Output the coordinates of the difference."
"Between the two images, find the true variation that exists in the second image. Return its location in bounding box format."
"Compare the pair of images. Where is the real and only difference in the second image? Provide the coordinates."
"Analyze the difference between these images. Identify and locate the actual modified region in the second image only."
**Task3**
"The object marked with a red bounding box in the first image (<|box_start|> (x_min, y_min), (x_max, y_max) <|box_end|>) is shared by these two images. Locate and identify it in the num image."
"In the first image, the object highlighted with a red bounding box (<|box_start|> (x_min, y_min), (x_max, y_max) <|box_end|>) is common to both images. Please recognize and locate it in the num image."
"The object outlined by a red bounding box in the first image (<|box_start|> (x_min, y_min), (x_max, y_max) <|box_end|>) appears in both images. Can you identify and find its position in the num image?"
"The object with a red bounding box in the first image (<|box_start|> (x_min, y_min), (x_max, y_max) <|box_end|>) is shared between these two images. Locate and recognize it in the num image."
"The object marked in red in the first image (<|box_start|> (x_min, y_min), (x_max, y_max) <|box_end|>) is common across both images. Find and identify it in the num image."
"The object highlighted by the red box in the first image (<|box_start|> (x_min, y_min), (x_max, y_max) <|box_end|>) is shared with the second image. Locate it in the num image and provide its position."
"Both images contain a common object marked with a red bounding box in the first image (<|box_start|> (x_min, y_min), (x_max, y_max) <|box_end|>). Find and identify this object in the num image."
"In the first image, the object marked by the red bounding box (<|box_start|> (x_min, y_min), (x_max, y_max) <|box_end|>) appears in both. Can you locate it in the num image?"
"The object in the first image, marked by a red bounding box (<|box_start|> (x_min, y_min), (x_max, y_max) <|box_end|>), is also in the second image. Identify and locate it in the num image."
"In the first image, the object enclosed by the red bounding box (<|box_start|> (x_min, y_min), (x_max, y_max) <|box_end|>) is the same as in the second image. Locate it in the num image and identify its position."

## Template for Instruction Data Generation

**Task4**

"In the first image, a red bounding box marks a specific object (<|box_start|> (x_min, y_min), (x_max, y_max) <|box_end|>). Your task is to identify and localize the same object in the num image."

"The object enclosed in red in the first image (<|box_start|> (x_min, y_min), (x_max, y_max) <|box_end|>) also appears in the num image. Detect and locate it accordingly."

"Focus on the object highlighted by the red box in the first image (<|box_start|> (x_min, y_min), (x_max, y_max) <|box_end|>). Find and mark this same object in the num image."

"Observe the red-boxed object in the first image (<|box_start|> (x_min, y_min), (x_max, y_max) <|box_end|>). Identify where it appears in the num image."

"The first image contains an object inside a red bounding box (<|box_start|> (x_min, y_min), (x_max, y_max) <|box_end|>). Detect this same object in the num image."

"An object is annotated with a red box in the first image (<|box_start|> (x_min, y_min), (x_max, y_max) <|box_end|>). Determine where the same object appears in the num image."

"Use the red-bounded object in the first image (<|box_start|> (x_min, y_min), (x_max, y_max) <|box_end|>) as a reference. Identify its location in the num image."

"Locate in the num image the object that corresponds to the red-marked region in the first image (<|box_start|> (x_min, y_min), (x_max, y_max) <|box_end|>)."

"The first image includes an object shown with a red bounding box (<|box_start|> (x_min, y_min), (x_max, y_max) <|box_end|>). Recognize and localize this same object in the num image."

"Refer to the red-outlined region in the first image (<|box_start|> (x_min, y_min), (x_max, y_max) <|box_end|>). Locate the corresponding object in the num image."

**Task5**

"Given image-1 and image-2, identify and localize the object from image-1 within image-2."

"Based on the object shown in image-1, determine its corresponding location in image-2."

"Observe the object in image-1. Where does it appear in image-2? Mark the location."

"Find the region in image-2 that corresponds to the object highlighted in image-1."

"Refer to image-1 and locate the same object in image-2."

"Your task is to recognize the object from image-1 and indicate where it is in image-2."

"Using image-1 as a reference, identify the location of the same object in image-2."

"Locate the counterpart of the object shown in image-1 within image-2."

"Match the object in image-1 to its corresponding region in image-2 and provide its location."

"Analyze the object in image-1 and find its equivalent presence in image-2 by marking its location."

**Task6**

"You are given a source image and several cropped regions. Identify where the num region belongs in the source image."

"Observe the original image and its cropped parts. Locate the num region in the source image."

"Given one complete image and multiple region crops, find where the num one fits in the original image."

"You are shown a source image and some regional cutouts. Point out where the num region comes from."

"Refer to the original image and determine the location of the num region shown afterward."

"Analyze the full image and match the num region image to its location within it."

"Based on the source image, indicate where the num region patch belongs."

"Here is a source image followed by cropped regions. Find the position of the num region in the source."

"You are given a full image and several region patches. Locate the num patch within the source image."

## Template for Instruction Data Generation

**Task7**
"You are given total images. Based on the red bounding box in the first image, locate the corresponding region in the num image that shares a similar function or meaning."
"Among the total provided images, examine the red-highlighted area in the first image and identify the region in the num image that matches it semantically or functionally."
"You are given total images. Consider the red-marked region in the first image. In the num image, find the area that best aligns with it in terms of purpose or meaning."
"From the total images below, determine which region in the num image corresponds to the red-boxed area in the first image."
"You are given total images. Study the red region in the first image. Then, in the num image, identify the location that serves a similar role or conveys a similar idea."
"You are given total images. Take a close look at the red-bounded area in the first image. Locate the corresponding region in the num image that reflects the same concept."
"You are given total images. Focus on the red box in the first image. Your task is to find the equivalent region in the num image that shares its function or meaning."
"You are given total images. Analyze the highlighted region in the first image. In the num image, point out the area that represents the same functional or semantic content."
"Given total images, compare the red-boxed area in the first image with the num image and find the corresponding part."
"You are given total images. Observe the first image where a red region is marked. Identify the most similar region in the num image in terms of functionality or semantics."

**Task8**
"Identify the bounding box of the region described by the following expression: <|object_ref_start|> object name <|object_ref_end|>."
"Locate the region corresponding to the following structure and provide its bounding box:<|object_ref_start|> object name <|object_ref_end|>."
"What is the bounding box for the region denoted by <|object_ref_start|> object name <|object_ref_end|>?"
"Provide the bounding box for the following entity mentioned in the image: <|object_ref_start|> object name <|object_ref_end|>."
"Identify and annotate the bounding box of <|object_ref_start|> object name <|object_ref_end|>."
"Indicate the bounding box of the area that corresponds to <|object_ref_start|> object name <|object_ref_end|>."
"Determine the coordinates of the bounding box for the target structure: <|object_ref_start|> object name <|object_ref_end|>."
"What is the bounding box for the region denoted by <|object_ref_start|> object name1 <|object_ref_end|> and <|object_ref_start|> object name2 <|object_ref_end|>?"

## C    Data statistics of MedSG-188K

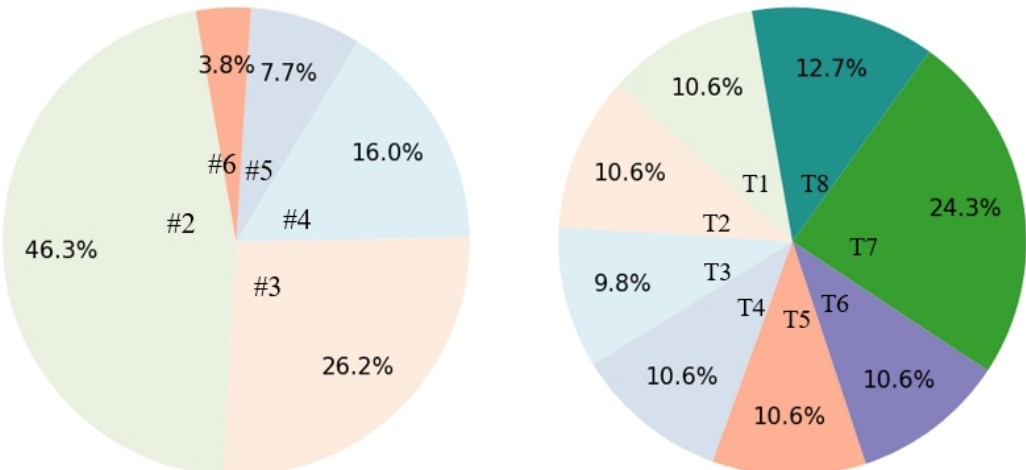

Figure 6: Proportions of image sequence length (**left**), data distribution across tasks (**right**) in MedSG-188K.

## D    Evaluation Metric

We evaluate model performance using two standard metrics: Intersection over Union (IoU) and Accuracy at IoU threshold 0.5 (Acc@0.5). These metrics are widely adopted in visual grounding to measure localization quality.

IoU quantifies the overlap between the predicted bounding box $B_{\text{pred}}$ and the ground-truth bounding box $B_{\text{gt}}$, and is defined as:

$$\text{IoU} = \frac{\text{Area}(B_{\text{pred}} \cap B_{\text{gt}})}{\text{Area}(B_{\text{pred}} \cup B_{\text{gt}})} \tag{1}$$

Acc@0.5 measures the proportion of predictions whose IoU with the ground truth exceeds 0.5. It is defined as:

$$\text{Acc@0.5} = \frac{1}{N} \sum_{i=1}^{N} \mathbb{I}(\text{IoU}_i \geq 0.5) \tag{2}$$

Here, $N$ is the total number of samples, and $\mathbb{I}(\cdot)$ is the indicator function that returns 1 if the condition is true, and 0 otherwise.

## E    Additional Analysis

### E.1    More Results

We benchmarked more MLLMs on MedSG-Bench, the results are summarized in Table 5. We grouped the targets by their bounding box area ratio into small (0–1%), medium (1–10%), and large (>10%), and evaluated model performance within each group, the results are summarized in Table 7.

### E.2    The potential bias of question generations

To examine whether the use of a single large language model (LLM) introduces bias in our question generation process, we conducted an additional comparative experiment across multiple LLMs, including GPT-4[10], Claude[45], and DeepSeek[46]. All generated questions were manually reviewed to ensure that they accurately preserved the original intent and complied with the standardized instruction format.

We then re-evaluated benchmarked models using questions generated by each LLM individually. The results, summarized in Table 6, report the average performance measured by IoU and acc@0.5. Despite minor variations among the three prompting settings, the observed differences remained within a narrow and acceptable range, suggesting that the core conclusions reported in the Main Results section are stable and robust to the choice of question generator.

Table 5: Performance of other MLLMs on MedSG-Bench. IDG: Image Difference Grounding; ICG: Image Consistency Grounding; RDG: Registered Difference Grounding; NRDG: Non-registered Difference Grounding; MV: Multi-view Grounding; OT: Object Tracking; VCG: Visual Concept Grounding; VPG: Visual Patch Grounding; CMG: Cross-modal Grounding; RG: Referring Grounding; Avg.: Average; IoU and acc@0.5 for all results are shown, all numbers are in percentages.

| Model | Size | IDG | | ICG | | | | | | Avg. |
| | | RDG | NRDG | MV | OT | VCG | VPG | CMG | RG | |
|---|---|---|---|---|---|---|---|---|---|---|
| **General-purpose MLLMs** | | | | | | | | | | |
| InternVL2[58] | 8B | 0.18 / 0.00 | 0.38 / 0.00 | 17.34 / 7.03 | 26.45 / 21.20 | 5.56 / 0.80 | 10.36 / 0.70 | 6.23 / 1.00 | 15.73 / 7.69 | 10.24 / 4.59 |
| InternVL2[58] | 76B | 0.15 / 0.00 | 0.15 / 0.00 | 10.00 / 3.90 | 15.56 / 11.80 | 3.39 / 0.40 | 6.64 / 1.10 | 2.83 / 0.75 | 15.69 / 9.92 | 6.88 / 3.53 |
| InternVL2.5[59] | 8B | 0.26 / 0.00 | 0.38 / 0.00 | 13.52 / 3.56 | 20.82 / 13.80 | 1.96 / 0.00 | 5.25 / 0.00 | 4.70 / 0.85 | 9.56 / 3.44 | 7.04 / 2.56 |
| InternVL2.5[59] | 78B | 0.24 / 0.10 | 0.32 / 0.10 | 9.16 / 2.08 | 16.18 / 10.00 | 4.32 / 0.50 | 11.86 / 2.30 | 5.48 / 1.25 | 10.67 / 4.52 | 7.29 / 2.55 |
| **Medical-domain specialized MLLMs** | | | | | | | | | | |
| LLaVA-Med v1.5[60] | 7B | 0.32 / 0.00 | 0.46 / 0.00 | 6.45 / 2.86 | 11.49 / 5.61 | 8.41 / 0.70 | 12.74 / 4.21 | 6.58 / 4.35 | 7.44 / 1.78 | 6.29 / 1.64 |
| BiMediX2[61] | 8B | 0.24 / 0.01 | 0.28 / 0.00 | 4.11 / 1.48 | 8.66 / 4.95 | 7.42 / 1.12 | 10.67 / 3.66 | 4.38 / 2.71 | 7.62 / 2.02 | 4.83 / 1.29 |

Table 6: Bias Analysis in Question Generation. IoU and acc@0.5 for all results are shown, all numbers are in percentages.

| Model | Size | Avg(GPT-4) | Avg(DeepSeek) | Avg(Claude) | Avg(Ori) |
|---|---|---|---|---|---|
| Qwen2.5-VL[11] | 3B | 10.51 / 3.86 | 10.60 / 3.85 | 10.31 / 9.02 | 10.94 / 4.20 |
| Qwen2.5-VL[11] | 7B | 11.25 / 15.29 | 10.87 / 4.13 | 11.19 / 4.41 | 12.31 / 4.90 |
| Qwen2.5-VL[11] | 72B | 13.45 / 6.37 | 13.35 / 6.29 | 13.39 / 6.41 | 13.35 / 6.12 |
| MiniCPM-V-2_6[49] | 8B | 12.72 / 4.59 | 13.33 / 5.13 | 12.61 / 4.30 | 13.24 / 5.27 |
| MiniCPM-O-2_6[50] | 8B | 10.68 / 3.85 | 10.34 / 3.51 | 10.27 / 3.32 | 10.12 / 3.23 |
| mPLUG-Owl3[51] | 7B | 10.92 / 2.86 | 10.71 / 2.69 | 11.04 / 2.92 | 13.22 / 3.19 |
| Mantis-Idefics2[52] | 8B | 10.33 / 4.35 | 10.02 / 4.07 | 10.06 / 3.91 | 9.90 / 3.91 |
| LLaVA-OneVision[53] | 7B | 13.55 / 5.51 | 11.59 / 3.44 | 12.46 / 3.47 | 12.39 / 3.47 |
| InternVL2.5[59] | 8B | 7.46 / 2.78 | 7.83 / 2.72 | 7.13 / 2.64 | 7.04 / 2.56 |
| Migician[28] | 7B | 20.31 / 11.39 | 20.53 / 11.91 | 20.43 / 11.46 | 20.29 / 11.39 |
| HuatuoGPT-Vision[12] | 7B | 9.08 / 2.71 | 9.20 / 2.59 | 9.18 / 2.41 | 8.97 / 2.36 |
| MedSeq-Grounder (Ours) | 7B | 72.68 / 79.98 | 71.67 / 78.76 | 72.86 / 80.18 | 72.55 / 79.71 |

Table 7: Fine-grained performance of different MLLMs on MedSG-Bench. IoU and acc@0.5 for all results are shown, all numbers are in percentages.

| Model | Size | Avg_small | Avg_medium | Avg_large |
|---|---|---|---|---|
| **General-purpose MLLMs** | | | | |
| Qwen2.5-VL[11] | 3B | 2.27 / 1.28 | 9.53 / 4.66 | 24.73 / 7.47 |
| Qwen2.5-VL[11] | 7B | 1.69 / 0.48 | 8.31 / 3.74 | 20.13 / 7.91 |
| Qwen2.5-VL[11] | 32B | 3.42 / 1.21 | 12.01 / 5.41 | 26.46 / 12.78 |
| Qwen2.5-VL[11] | 72B | 3.82 / 1.18 | 11.96 / 5.31 | 25.92 / 13.25 |
| MiniCPM-V-2_6[49] | 8B | 2.93 / 0.55 | 15.35 / 6.40 | 24.93 / 10.36 |
| MiniCPM-O-2_6[50] | 8B | 2.60 / 0.38 | 10.93 / 3.25 | 19.82 / 7.36 |
| mPLUG-Owl3[51] | 7B | 2.41 / 0.00 | 14.57 / 22.75 | 26.86 / 8.54 |
| Mantis-Idefics2[52] | 8B | 2.92 / 1.10 | 14.47 / 6.72 | 12.74 / 3.51 |
| LLaVA-OneVision[53] | 7B | 1.19 / 0.00 | 15.33 / 4.28 | 23.51 / 7.23 |
| LLaVA-OneVision[53] | 72B | 3.27 / 0.55 | 14.68 / 3.99 | 25.38 / 13.87 |
| InternVL2[58] | 8B | 2.24 / 1.01 | 12.64 / 6.09 | 18.09 / 7.40 |
| InternVL2[58] | 76B | 1.12 / 0.32 | 7.46 / 3.49 | 14.39 / 8.29 |
| InternVL2.5[59] | 8B | 2.06 / 0.55 | 10.24 / 4.41 | 9.17 / 2.54 |
| InternVL2.5[59] | 78B | 1.33 / 0.38 | 8.64 / 2.86 | 13.85 / 5.25 |
| InternVL3[54] | 8B | 1.81 / 0.32 | 9.06 / 2.49 | 20.49 / 8.50 |
| InternVL3[54] | 14B | 2.23 / 0.52 | 12.76 / 5.12 | 19.09 / 8.97 |
| InternVL3[54] | 38B | 2.47 / 0.61 | 11.12 / 4.70 | 20.71 / 9.65 |
| InternVL3[54] | 78B | 1.26 / 0.38 | 6.80 / 2.57 | 13.46 / 7.11 |
| Migician[28] | 7B | 11.24 / 4.72 | 22.05 / 11.89 | 30.69 / 20.36 |
| **Medical-domain specialized MLLMs** | | | | |
| HuatuoGPT-Vision[12] | 7B | 2.55 / 0.35 | 11.55 / 3.88 | 14.20 / 2.83 |
| HuatuoGPT-Vision[12] | 34B | 2.90 / 0.43 | 11.02 / 3.38 | 12.93 / 2.41 |
| MedSeq-Grounder (Ours) | 7B | 68.37 / 75.83 | 69.32 / 75.26 | 83.88 / 92.55 |

## E.3 Effect of clinical windowing on model performance

To investigate whether different window settings influence model performance, we conducted additional experiments under two settings: (1) applying only min–max normalization, and (2) applying clinical windowing followed by min–max normalization.

These experiments are conducted on CT datasets, including AbdomenCT1K, LUNA16, and COVID-19-20, where window settings are clinically significant and can substantially affect the visibility of anatomical structures. Specifically, we adopted organ-specific window ranges as follows: Lung: [-600, 1500]; Liver: [60, 150]; Spleen: [60, 150]; Kidney: [40, 400]. The results, summarized in Table 8, report the IoU and acc@0.5 on Registered Difference Grounding, Visual Concept Grounding, and Visual Patch Grounding tasks.

From the results, we observed that clinical windowing led to perfomance gains for models with strong visual perception capabilities (e.g., Migician) or prior exposure to medical data (e.g., HuatuoGPT-Vision), with our proposed MedSeq-Grounder achieving the most significant improvement. We also find that most general-purpose MLLMs typically suffered performance drops under the same setting. This pattern suggests that clinical windowing introduces distribution shifts that challenge general models, while models equipped with robust perceptual abilities and domain-specific knowledge can leverage enhanced contrast and localized visual cues more effectively.

Table 8: Evaluation Results with clinical windowing and min-max normalization on CT datasets. RDG: Registered Difference Grounding; VCG: Visual Concept Grounding; VPG: Visual Patch Grounding; Avg.: Average; IoU and acc@0.5 for all results are shown, all numbers are in percentages.

| Model | Size | RDG | | VCG | | VPG | | Avg | |
|---|---|---|---|---|---|---|---|---|---|
| | | ori | window | ori | window | ori | window | ori | window |
| Qwen2.5-VL[11] | 3B | 0.31 0.00 | 0.29 0.00 | 11.81 1.00 | 8.87 0.25 | 28.23 2.93 | 26.17 1.95 | 11.67 1.13 | 10.23 0.64 |
| Qwen2.5-VL[11] | 7B | 1.15 0.17 | 0.59 0.00 | 9.73 1.00 | 5.98 1.00 | 26.37 3.41 | 29.21 4.63 | 10.90 1.34 | 10.42 1.63 |
| Qwen2.5-VL[11] | 72B | 3.31 2.32 | 3.08 1.33 | 13.59 4.50 | 10.11 2.50 | 28.08 6.34 | 27.54 6.10 | 13.41 4.10 | 12.17 3.04 |
| MiniCPM-V-2_6[49] | 8B | 1.25 0.00 | 1.33 0.00 | 14.18 2.25 | 12.64 3.25 | 29.46 13.90 | 29.97 12.93 | 13.09 4.67 | 12.84 4.67 |
| MiniCPM-O-2_6[50] | 8B | 1.55 0.00 | 1.64 0.00 | 12.26 1.50 | 13.34 1.50 | 25.27 11.46 | 23.60 10.00 | 11.46 3.75 | 11.35 3.33 |
| Mantis-Idefics2[52] | 8B | 0.08 0.00 | 0.09 0.00 | 10.24 0.75 | 9.01 0.25 | 11.03 0.49 | 9.73 0.73 | 6.13 0.35 | 5.41 0.28 |
| LLaVA-OneVision[53] | 7B | 1.32 0.00 | 1.02 0.00 | 13.45 1.00 | 15.28 1.25 | 21.96 6.34 | 20.98 3.17 | 10.74 2.12 | 10.85 1.27 |
| InternVL2.5[59] | 8B | 0.14 0.00 | 0.15 0.00 | 1.67 0.00 | 1.06 0.00 | 5.01 0.00 | 4.44 0.00 | 1.99 0.00 | 1.65 0.00 |
| Migician[28] | 7B | 10.06 5.31 | 16.97 8.29 | 16.87 6.75 | 12.65 5.25 | 21.73 6.10 | 25.94 11.22 | 15.37 5.94 | 18.35 8.28 |
| HuatuoGPT-Vision[12] | 7B | 1.24 0.17 | 1.41 0.00 | 6.85 0.00 | 7.95 0.50 | 12.89 0.73 | 14.80 1.95 | 6.21 0.28 | 7.15 0.71 |
| MedSeq-Grounder (Ours) | 7B | 82.56 92.04 | 86.84 96.68 | 65.54 70.75 | 89.23 94.50 | 80.54 94.39 | 88.01 100.00 | 77.15 86.69 | 87.86 97.03 |

## E.4 Failure case study

We conducted a detailed failure analysis to better understand model behavior. Our initial observations show that the most models are able to correctly follow instructions and output coordinates in the required format.

However, as the visual context becomes more complex, model performance drops significantly. For example, Qwen2.5-VL frequently produces bounding boxes that span nearly the entire image ([0, 0, x_max, y_max]), and InternVL3 often outputs predictions that are spatially misaligned with the target region, as illustrated in Fig. 7.

# F  Potential negative societal impacts

While the proposed benchmark includes eight tasks spanning medical image sequences, the resulting performance is intended for reference purposes only. High scores achieved by MLLMs on MedSG-Bench do not necessarily indicate clinical readiness or real-world applicability. Any deployment in clinical settings requires thorough validation and oversight from qualified medical professionals to ensure safety and reliability.

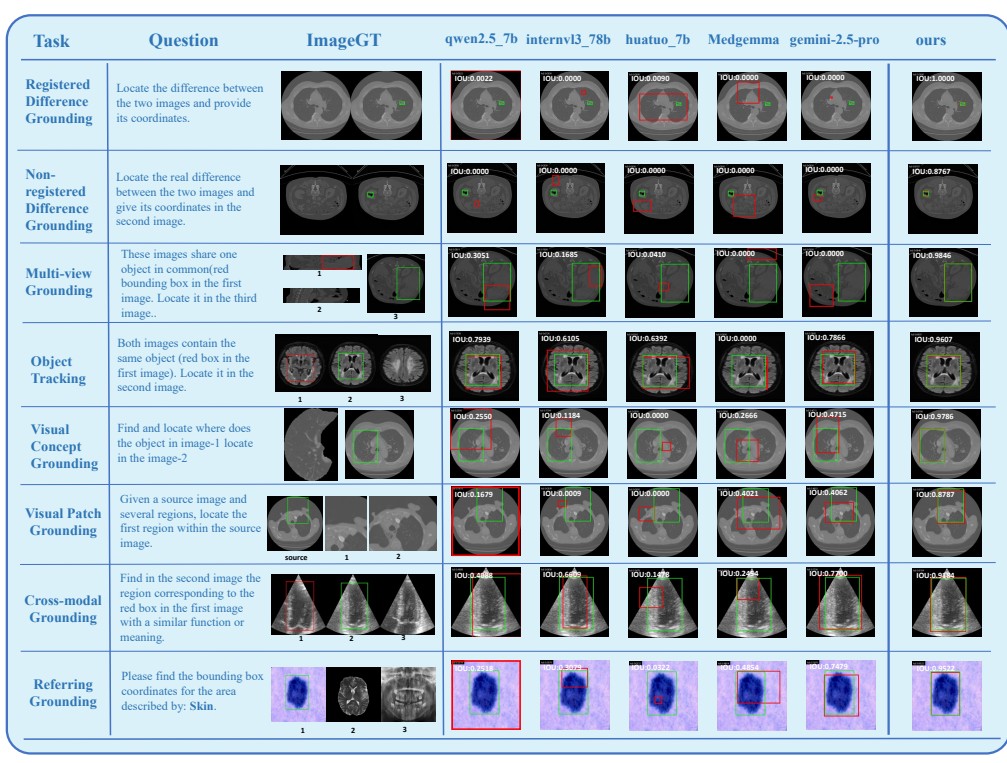

Figure 7: Visualization of samples in MedSG-Bench.

