# OpenReview forum: "MedSG-Bench: A Benchmark for Medical Image Sequences Grounding"
_NeurIPS.cc/2025/Datasets_and_Benchmarks_Track — NeurIPS 2025 Datasets and Benchmarks Track spotlight_

### Official Review · Reviewer_QurD · 2025-06-06

**Rating:** 5
**Confidence:** 4

**Summary:**

This paper introduces MedSG-Bench, the first comprehensive benchmark for medical image sequences grounding, addressing a critical gap in evaluating multimodal large language models (MLLMs) on sequential medical images. The benchmark comprises 8 VQA-style tasks organized into Image Difference Grounding and Image Consistency Grounding paradigms, constructed from 76 medical datasets across 10 imaging modalities with 9,630 question-answer pairs. The authors evaluate 20+ state-of-the-art MLLMs and develop MedSeq-Grounder, revealing significant limitations in current models for medical sequence understanding.

**Dataset Code Accessibility:**

Yes

**Ethical Considerations:**

No, there are no or only very minor ethics concerns

**Final Justification:**

The author solved all my questions, I maintain my rating of Accept (5).

**Limitations Weaknesses:**

1. The paper said "apply min-max normalization (Section 3.1.2) following [30]", but for DICOM images, clinicians use different window settings to visualize different anatomical structures. This preprocessing may alter the clinical interpretation of images and could explain the universally poor model performance.

2. Lacks detailed information about how they prompt the models during evaluation. While their dataset uses (x_min, y_min), (x_max, y_max) format, baseline models may have been trained with different formats (center+width/height, normalized coordinates, different notation). Will this be the reason for the baseline model's poor performance?

3. The paper needs a deeper investigation of why models fail. Can current models produce accurate coordinates even for simple cases with provided blue bounding boxes? Understanding whether failures stem from coordinate prediction, visual understanding, or instruction following would provide actionable insights beyond simply documenting poor performance.

4. Not necessary, given the importance of grounding tasks, it would be great if the authors could benchmark commercial api models like o1/o3 with visual capabilities. These models might demonstrate significantly different performance patterns.

**Strengths Contributions:**

In General, the paper is well-motivated,  provides clear task definitions, detailed statistics, and promises full resource release.

1. The paper addresses an underexplored but critical area in medical AI. While existing benchmarks focus on single-image scenarios, real clinical workflows inherently require sequential image analysis (pre-/post-treatment comparisons, disease progression tracking). This is well-motivated.

2. The 8-task framework (Section 3.2, Figure 3) is thoughtfully designed, covering difference tasks from basic difference detection to complex cross-modal reasoning.

3. The construction from 76 datasets (Table 4) with proper licensing considerations demonstrates thorough effort.

---

> ### Author Rebuttal · Authors · 2025-07-31
>
> We sincerely thank the reviewer for the thoughtful insights and valuable feedback. We are encouraged to see that the reviewer believes our work is "well motivated" and our benchmark is "thoughtfully designed". Below, we address each of the concern raised by the reviewer.
>
> **W1: "apply min-max normalization and different clinical window settings"**
>
> We thank the reviewer for raising such an interesting and important question.
>
> (1) To invetigate whether different window sizes affects model performance, we have included additional experiments under two settings: 1) applying only min-max normalization, and 2) applying clinical windowing followed by min-max normalization.
>
> The above experiments are conducted on CT data (including AbdomenCT-1K, LUNA16, and COVID-19-20 datasets), as window settings are clinically significant and particularly affect the visibility of anatomical structures in CT modalities. We applied corresponding organ-specific window settings as follows: Lung: \[-600, 1500\], Liver: \[60,150\], Spleen: \[60,150\], Kidney: \[40,400\]. The result (IoU (acc\@0.5)) on Registered Difference Grounding, Visual Concept Grounding, and Visual Patch Grounding are summarized below:
>
> ||Size| RDG_ori| RDG_window| VCG_ori|VCG_window| VPG_ori|VPG_window|Avg_ori|Avg_window|
> | ------ |:-:| ------ | ------ | ------ | ------ | ------ | ------ | ------ | ------ |
> |Qwen2.5-VL|3B|0.31 (0.00)| 0.29 (0.00)|11.81 (1.00)|8.87 (0.25)| 28.23 (2.93)|26.17 (1.95)|11.67 (1.13)|10.23 (0.64)|
> |Qwen2.5-VL|7B|1.15 (0.17)|0.59 (0.00)|9.73 (1.00)|5.98 (1.00)|26.37 (3.41)|29.21 (4.63)|10.90 (1.34)|10.42 (1.63)
> |Qwen2.5-VL|72B|3.31 (2.32)|3.08 (1.33)|13.59 (4.50)|10.11 (2.50)|28.08 (6.34)|27.54 (6.10)|13.41 (4.10)|12.17 (3.04)
> | MiniCPM-V-2_6 |8B|1.25 (0.00)|1.33 (0.00)|14.18 (2.25)|12.64 (3.25)|29.46 (13.90)|29.97 (12.93)|13.09 (4.67)|12.84 (4.67)
> | MiniCPM-O-2_6 |8B|1.55 (0.00)|1.64 (0.00)|12.26 (1.50)|13.34 (1.50)|25.27 (11.46)|23.60 (10.00)|11.46 (3.75)|11.35 (3.33)
> | Mantis-Idefics2 |8B|0.08 (0.00)|0.09 (0.00)|10.24 (0.75)|9.01 (0.25)|11.03 (0.49)|9.73 (0.73)|6.13 (0.35)|5.41 (0.28)
> | LLaVA-OneVision |7B|1.32 (0.00)|1.02 (0.00)|13.45 (1.00)|15.28 (1.25)|21.96 (6.34)|20.98 (3.17)|10.74 (2.12)|10.85 (1.27)
> |InternVL2.5 |8B|0.14 (0.00)|0.15 (0.00)|1.67 (0.00)|1.06 (0.00)|5.01 (0.00)|4.44 (0.00)|1.99 (0.00)|1.65 (0.00)
> |Migician|7B|10.06 (5.31)|16.97 (8.29)|16.87 (6.75)|12.65 (5.25)|21.73 (6.10)|25.94 (11.22)|15.37 (5.94)|18.35 (8.28)
> |HuatuoGPT-Vision|7B|1.24 (0.17)|1.41 (0.00)|6.85 (0.00)|7.95 (0.50)|12.89 (0.73)|14.80 (1.95)|6.21 (0.28)|7.15 (0.71)
> |MedSeq-Grounder|7B|82.56 (92.04)|86.84 (96.68)|65.54 (70.75)|89.23 (94.50)|80.54 (94.39)|88.01 (100.00)|77.15 (86.69)|87.86 (97.03)
>
> (2) From the results, we observed that clinical windowing led to perfomance gains for models with strong visual perception capabilities (e.g., Migician) or prior exposure to medical data (e.g., HuatuoGPT-Vision), with our proposed MedSeq-Grounder achieving the most significant improvement. We also find that most general-purpose MLLMs typically suffered performance drops under the same setting. This pattern suggests that clinical windowing introduces distribution shifts that challenge general models, while models equipped with robust perceptual abilities and domain-specific knowledge can leverage enhanced contrast and localized visual cues more effectively.
>
> **W2: "Lack detailed information about how they prompt the models during evaluation"**
>
> We thank the reviewer for pointing out this subtle yet important aspect of the evaluation process. We fully agree that inconsistencies in coordinate formats across different models could impact the evaluation outcomes, and **we had already taken this into account during our initial evaluation design.**
>
> Specifically, for each baseline model, we consulted the official documents or papers to understand the expected coordinate format. For example, the InternVL series models normalizes coordinates to the range \[0, 1000\], while Qwen2.5-VL support absolute coordinates output, and Gemini 2.5 Pro uses the format \[y_min, x_min, y_max, x_max\].
>
> Accordingly, before passing questions into the models, we convert all bounding box coordinates into the required format supported by the target model. After obtaining the model's predictions, we apply an inverse transformation to restore the coordinates to the standard absolute format \[x_min, y_min, x_max, y_max\] format, which is used to compute the IoU metric against our ground truth annotations.
>
> **W3: "The paper needs a deeper investigation of why models fail"**
>
> We thank the reviewer for insightful suggestions. In response, we conducted a detailed failure analysis to better understand model behavior.
>
> (1) Our initial observations show that **most models are able to correctly follow instructions and output coordinates in the required format.** In relatively simple cases, such as images containing a single large lesion with clear boundaries, the predicted bounding boxes are generally accurate and often achieve an IoU greater than 0.7.
>
> (2) However, as the visual context becomes more complex, model performance drops significantly. For example, Qwen2.5-VL frequently produces bounding boxes that span nearly the entire image (\[0, 0, x_max, y_max\]), and InternVL3 often outputs predictions that are spatially misaligned with the target region.
>
> (3) These findings suggest that the primary limitation lies not in instruction-following or coordinate formatting, but in the lack of robust multi-image visual understanding in medical contexts.
>
> To further investigate this hypothesis, we conducted an experiment using the RDG task on the JSRT dataset under the following three settings: (1) fine-tuning only the LLM decoder, (2) fine-tuning only the vision encoder, and (3) fine-tuning both the vision encoder and the projection layer. The results are summarized below.
>
> The results show that tuning the vision encoder yields the most substantial performance improvement, highlighting **the critical role of robust visual understanding in medical image sequence grounding**.
>
> ||epoch 0 |epoch 1| epoch 2|epoch 3|epoch 4|epoch 5|epoch 6|epoch 7|epoch 8|epoch 9|
> | ------ | ------ | ------ | ------ | ------ | ------ | ------ | ------ | ------ |:-:|:-:|
> | LLM Decoder|0.19|0.31|0.51|0.69|0.95|0.94|1.57|1.38|1.11|1.39|
> | Vision Encoder|2.42|34.68|50.42|57.73|58.47|62.64|65.49|63.28|65.94|64.66|
> |Vision Encoder+projector| 2.01|27.16|52.24|47.81|47.58|71.88|70.71|74.18|78.97|78.75|
>
>
> **W4: "Not necessary, it would be great if the authors could benchmark commercial api models"**
>
> We appreciate the reviewer's valuable suggestion. We have additionally included three mainstream proprietary MLLMs: **GPT-4o, Claude Sonnet 4, and Gemini 2.5 Pro**. Corresponding results (IoU(acc\@0.5)) are summarized below.
> Notably, Gemini 2.5 Pro achieved the highest overall performance among all the benchmarked models, including those reported in Table 3 of the manuscript. We indeed observed different performance patterns. Notably, proprietary models performed well on Visual Patch Grounding and Referring Grounding, indicating stronger fine-grained visual perception and cross-modal understanding. This aligns with the reviewer’s expectation and confirms the value of including such models.
>
> Model| RDG| NRDG| MV|OT|VCG| VPG| CMG|RG|Avg|
> | ------ | ------ | ------ | ------ | ------ | ------ | ------ | ------ | ------ |:-:|
> |GPT-4o |2.42 (0.40)|3.45 (0.20)|16.51 (8.62)|28.19 (23.90)|13.18 (4.70)|38.05 (26.40)|16.02 (4.95)|23.08 (18.02)|17.70 (10.60)|
> |Claude Sonnet 4|0.67 (0.00)|0.81 (0.10)|12.56 (3.57)|23.11 (16.50)|6.93 (1.40)|27.44 (13.80)|9.04 (1.80)|19.57 (10.80)|12.51 (5.76)|
> |Gemini 2.5 Pro|9.36 (3.20)|7.29 (2.00)|14.26 (6.71)|19.32 (13.80)|14.94 (10.70)|41.11 (49.20)|24.44 (28.12)|28.12 (22.67)|20.66 (15.61)|

---

> > ### Comment · Reviewer_QurD · 2025-08-03
> >
> > Thank you for the comprehensive rebuttal. Your additional experiments and analysis have addressed all my concerns. I maintain my rating of Accept (5).

---

> > > ### Author Response · Authors · 2025-08-04
> > > **Official Comment by Authors**
> > >
> > > We are really glad that our responses address your concerns! Thanks so much for your thoughtful questions and insightful discussion. We will carefully include our discussion in the final version. We deeply thank the effort you made during the whole process!

---

### Official Review · Reviewer_ejM7 · 2025-06-30

**Rating:** 4
**Confidence:** 4

**Summary:**

The paper proposes a benchmark for visual grounding in MLLMs under sequential image scenarios. MedSG-Bench covers 76 datasets, 10 imaging modalities, and 8 VQA-style tasks. Additionally, the authors construct a large-scale instruction-tuning dataset, MedSG-188K, and develop an MLLM, MedSeq-Grounder, for fine-grained understanding.

**Dataset Code Accessibility:**

Yes

**Dataset Code Comments:**

Yes, this study has provided dataset and code links.

**Ethical Considerations:**

No, there are no or only very minor ethics concerns

**Final Justification:**

The rebuttals have addressed most of my concerns. Therefore, I raise the score to 'Borderline accept'.

**Limitations Weaknesses:**

1. The use of synthetic perturbations to simulate clinical variations is understandable. Authors should discuss the potential influence this design may have on the model's grounding behavior and provide justification for its validity.
2. Authors conduct extensive experiments with general-purpose MLLMs, yet only include a single model specialized in the medical domain. Based on this limited comparison, the conclusion that “medical-domain specialized models are often worse than general-purpose models” appears insufficiently supported.
3. Some figures in the paper appear to repeat information already presented in tables, leading to unnecessary redundancy.
4. More VQA results are recommended to be included in the supplementary materials to enhance the completeness of the evaluation.
5. The description of the MedSG-188K dataset lacks sufficient detail in paper. In particular, the authors do not clearly specify the number of images included in the newly constructed dataset.

**Strengths Contributions:**

1. This paper is the first to focus on the task of grounding in medical image sequences.
2. The paper presents eight well-designed tasks to systematically evaluate the performance of MLLMs.
3. The paper conducts experiments on various MLLMs, and all associated resources have been made publicly available.

---

> ### Author Rebuttal · Authors · 2025-07-31
>
> We sincerely thank the reviewer for acknowledging the novelty of our work, and for the thoughtful and constructive comments.
>
> **W1: "The potential influence of synthetic perturbations"**
>
> We thank the reviewer for raising this important question. We would like to clarify as follows:
>
> (1) In our work, synthetic pertubations are employed in the IDG task (as discussed in Section 3.2.1), which is designed to simulate clinically meaningful variations such as pre-/post-treatment comparisons. Given the scarcity of high-quality longitudinal data with fine-grained annotations in real-world clinical practice, prior studies \[1,2\] have demonstrated that such synthetic strategies can effectively enhance model's generalization.
>
> (2) To analyse the potential influence of this design, we have further curated a tiny real-world dataset comprising pre-/post-treatment CT images collected from a hospital, including six lung cancer and three mediastinal lymphoma cases. The benchmarked models and our MedSeq-Grounder are evaluated on this dataset (Corresponding results are shown below for your convenience). Results show that MedSeq-Grounder trained on synthetic perturbation data based on Qwen2.5-VL-7B, exhibited consistent and accurate grounding behavior, demonstrating that **the use of synthetic perturbations is an effective and clinically meaningful strategy for modeling real-world changes.** The tiny dataset will be released.
>
> |Model |Size|Avg_IoU|
> | ------ | ------ | ------ |
> |GPT-4o| --|3.45 |
> | Claude Sonnet 4| --|0.81|
> |Gemini 2.5 Pro| --|7.29|
> | Qwen2.5-VL|3B | 1.62|
> | Qwen2.5-VL|7B |1.25|
> | Qwen2.5-VL|72B|3.46|
> |MiniCPM-v-2_6 |8B|1.50|
> | MiniCPM-o-2_6|8B|1.63 |
> | Mantis-Idefics2 |8B| 0.62|
> |LLaVA-OneVision|7B|0.01
> |InternVL2.5 |8B|0.26
> |Migician|7B|16.72
> |HuatuoGPT-Vision|7B|8.04
> |MedSeq-Grounder|7B|54.34
>
> **W2: "Only include a single model specialized in the medical domain"**
>
> We appreciate the reviewer's valuable suggestion.
>
> (1) In response, we have extended our evaluation to include three additional publicly available medical-domain MLLMs: **MedGemma, LLaVA-Med v1.5, and BiMediX2**, to provide a more comprehensive evaluation on our MedSG-Bench. The corresponding results (IoU(acc\@0.5)) are summarized below.
>
> (2) We compared these results with those in Table 3 of the original manuscript. Consistently, the general-purpose MLLMs still outperform most medical-domain specialized models across various tasks, which further support our observation that medical-domain specialized models, despite being tailored for clinical applications, may not yet generalize as effectively as leading general-purpose models under grounding tasks in medical image sequences.
>
> |Model| RDG| NRDG| MV|OT|VCG| VPG| CMG|RG|Avg|
> | ------ | ------ | ------ | ------ | ------ | ------ | ------ | ------ | ------ |:-:|
> |MedGemma |0.45 (0.00)|0.84 (0.00)|7.80 (4.53)|26.82 (22.40)|11.31 (0.90)|26.59 (15.40)|5.92 (0.50)|10.01 (1.01)|10.55 (4.82)|
> |LLaVA Med v1.5|0.32 (0.00)|0.46 (0.00)|6.45 (2.86)|11.49 (5.61)|8.41 (0.70)|12.74 (4.21)|6.58 (4.35)|7.44 (1.78)|6.29 (1.64)|
> |BiMediX2|0.24 (0.01)|0.28 (0.00)|4.11 (1.48)|8.66 (4.95)|7.42 (1.12)|10.67 (3.66)|4.38 (2.71)|7.62 (2.02)|4.83 (1.29)|
> |HuatuoGPT-Vision-7B|1.35 (0.00)|1.84 (0.20)|10.42 (2.78)|14.57 (9.20)|7.99 (0.80)|15.52 (2.30)|9.46 (2.15)|9.60 (1.82)|8.97 (2.36)
> |HuatuoGPT-Vision-34B|1.44 (0.00)|2.15 (0.00)|9.41 (1.65)|13.25 (8.30)|6.43 (0.70)|14.53 (1.40)|10.60 (2.60)|8.60 (1.75)|8.57 (2.09)
> |MedSeq-Grounder|83.29 (93.20)|83.72 (94.10)|55.03 (60.19)|62.10 (67.20)|74.11 (82.60)|85.25 (98.80)|78.77 (82.75)|60.43 (65.59)|72.55 (79.71)
>
> **W3: "Some figures in the paper appear to repeat information already presented in tables, leading to unnecessary redundancy"**
>
> We thank the reviewer for raising this concern. Upon careful review, we have revised the figures and tables in the manuscript to ensure that they serve distinct and complementary purposes, avoiding any unnecessary redundancy. The updated captions (legends) are provided below.
>
> || Captions|
> | ------ | ------ |
> |Figure 1| Some examples of medical image sequence grounding.|
> |Figure 2|Visualization of MLLM performance on MedSG-Bench.|
> |Figure 3| An illustration of medical image sequences grounding tasks included in MedSG-Bench.|
> |Figure 4| Overview of the MedSG-Bench construction pipeline.|
> |Figure 5|Distribution characteristics of MedSG-Bench, including sequence length distribution across all samples, and target-to-image size ratio distribution.|
> |Figure 6| Distribution characteristics of MedSG-188K, including sequence length distribution across all samples, and target-to-image size ratio distribution.|
> |Table 1|Comparison with existing medical benchmarks.|
> |Table 2|Task-level statistics of MedSG-Bench, including the number of datasets, modalities, clinical tasks, maximum sequence length, and the total number of samples per task. |
> |Table 3|Quantitative performance of different MLLMs on MedSG-Bench.
> |Table 4|Detailed datasets information in MedSG-Bench.
>
> **W4: "More VQA results are recommended to be included in the supplementary materials"**
>
> We thank the reviewer for the valuable comment. We have extended the evaluation to include more VQA results to enhance the comprehensiveness of our experiments. Specifically,
>
> **(1) Results of More Models:** We expanded our evaluation by incorporating additional medical-domain MLLMs (**MedGemma, LLaVA-Med v1.5, and BiMediX2**, see W2 for details), as well as three mainstream proprietary models (**GPT-4o, Claude Sonnet 4, and Gemini 2.5 Pro**). The performance results are summarized below.
>
> Model|RDG|NRDG|MV|OT|VCG|VPG|CMG|RG|Avg|
> | ------ | ------ | ------ | ------ | ------ | ------ | ------ | ------ | ------ |:-:|
> |GPT-4o |2.42 (0.40)|3.45 (0.20)|16.51 (8.62)|28.19 (23.90)|13.18 (4.70)|38.05 (26.40)|16.02 (4.95)|23.08 (18.02)|17.70 (10.60)|
> |Claude Sonnet 4|0.67 (0.00)|0.81 (0.10)|12.56 (3.57)|23.11 (16.50)|6.93 (1.40)|27.44 (13.80)|9.04 (1.80)|19.57 (10.80)|12.51 (5.76)|
> |Gemini 2.5 Pro|9.36 (3.20)|7.29 (2.00)|14.26 (6.71)|19.32 (13.80)|14.94 (10.70)|41.11 (49.20)|24.44 (28.12)|28.12 (22.67)|20.66 (15.61)|
>
> **(2) Fine-Grained Results:** As **small-target** recognition is a critical challenge in the medical domain, we further categorized the targets into three groups based on their bounding box area ratio: small (0-1%), medium (1-10%), and large (>10%). We then evaluated model performance for each group. The corresponding per-task results will be included in the revised supplementary materials accordingly. In addition, we also provided the average performance (IoU(acc\@0.5)) below for your convenience.
> The results demonstrate that most MLLMs exhibit substantially reduced performance on small targets, underscoring their limitations in precise medical sequential grounding. In contrast, MedSeq-Grounder consistently achieves strong performance across all target sizes, demonstrating its robustness grounding capability in clinically challenging scenarios.
>
> |Model|Size| Avg_small |Avg_medium| Avg_large|
> | ------ |:-:| ------ | ------ | ------ |
> | Qwen2.5-VL |3B|2.27 (1.28)|9.53 (4.66)|24.73 (7.47)|
> | Qwen2.5-VL |7B|1.69 (0.48)|8.31 (3.74)|20.13 (7.91)|
> |Qwen2.5-VL|32B|3.42 (1.21)|12.01 (5.41)|26.46 (12.78)
> | Qwen2.5-VL |72B|3.82 (1.18)|11.96 (5.31)|25.92 (13.25)|
> | MiniCPM-V-2_6 |8B|2.93 (0.55)|15.35 (6.40)|24.93 (10.36)|
> | MiniCPM-O-2_6 |8B|2.60 (0.38)|10.93 (3.25)|19.82 (7.36)|
> | mPLUG-Owl3 |7B|2.41 (0.00)| 14.57 (22.75)|26.86 (8.54)|
> | Mantis-Idefics2 |8B|2.92 (1.10)|14.47 (6.72)|12.74 (3.51)|
> | LLaVA-OneVision |7B|1.19 (0.00)| 15.33 (4.28)|23.51(7.23)|
> |LLaVA-OneVision |72B|3.27 (0.55)|14.68 (3.99)|25.38 (13.87)
> |InternVL2|8B|2.24 (1.01)|12.64 (6.09)|18.09 (7.40)
> |InternVL2|76B|1.12 (0.32)|7.46 (3.49)|14.39 (8.29)
> | InternVL2.5 |8B|2.06 (0.55)|10.24 (4.41)| 9.17 (2.54)|
> |InternVL2.5|78B|1.33 (0.38)|8.64 (2.86)|13.85 (5.25)
> |InternVL3|8B|1.81 (0.32)|9.06 (2.49)|20.49 (8.50)
> |InternVL3|14B|2.23 (0.52)|12.76 (5.12)|19.09 (8.97)
> |InternVL3|38B|2.47 (0.61)|11.12 (4.70)|20.71 (9.65)
> |InternVL3|78B|1.26 (0.38)|6.80 (2.57)|13.46 (7.11)
> |Migician|7B|11.24 (4.72)|22.05 (11.89)|30.69 (20.36)|
> |HuatuoGPT-Vision|7B|2.55 (0.35)|11.55 (3.88)|14.20 (2.83)|
> |HuatuoGPT-Vision|34B|2.90 (0.43)|11.02 (3.38)|12.93 (2.41)
> |MedSeq-Grounder|7B|68.37 (75.83)|69.32 (75.26)|83.88 (92.55)|
>
> **W5: "The description of the MedSG-188K dataset lacks sufficient detail in paper"**
>
> Thanks for pointing this out.
> The detailed description of the MedSG-188K is as follows: MedSG-188K is a grounding instruction-tuning dataset constructed by collecting and unifying data from 76 publicly available medical datasets. It covers 10 imaging modalities and 114 clinical tasks, with a maximum of image sequence length of six. The distribution of sequence lengths and data volume is summarized in Figure 6 of the appendix. **In total, MedSG-188K comprises 324,359 images, and MedSG-Bench includes 24,341 images.** We will include this detailed information in the revised manuscript, and also presented in the table below for your convenience.
>
> | | CBCT| CT| CTA|Colonoscopy | Dermoscopy |Endoscopy|Fundus|MRI|US|X-ray|
> | ------ | ------ | ------ | ------ | ------ | ------ | ------ | ------ | ------ |:-:|:-:|
> | MedSG-Bench| 207| 13,448| 120|174| 249| 272| 73|8,769|415|614|
> | MedSG-188K|387 |187,556|1,659|1,340|634|4,647|686|120,888|3,242|3,320|
>
> | | Head & Neck | Thorax| Abdomen| Pelvis|Skin|
> | ------ | ------ | ------ | ------ | ------ | ------ |
> | MedSG-Bench| 6,685| 8,598| 7,843| 977| 238|
> | MedSG-188K| 104,660|107,015|100,904|11,146|634|
>
> **References**
>
> \[1\] Sun Y, Tan W, Gu Z, et al. A data-efficient strategy for building high-performing medical foundation models[J]. Nature Biomedical Engineering, 2025: 1-13.
>
> \[2\] Wu L, Zhuang J, Zhou Y, et al. Freetumor: Large-scale generative tumor synthesis in computed tomography images for improving tumor recognition[J]. arXiv preprint arXiv:2502.18519, 2025.

---

> > ### Comment · Reviewer_ejM7 · 2025-08-04
> > **Good Rebuttal**
> >
> > The authors' response and additional experiments have addressed my main concerns. The dataset presented is a valuable contribution to the field. I would still recommend reformatting Figure 5 to improve its readability. Based on these considerations, I am updating my review decision from 'borderline reject' to 'borderline accept'.

---

> > > ### Author Response · Authors · 2025-08-04
> > > **Official Comment by Authors**
> > >
> > > We are glad to hear that our response and additional experiments have addressed the reviewer’s concerns, and we sincerely appreciate the recognition of the value our work brings to the field. As suggested, we will revise the caption and formatting of Figure 5 in the final version to further improve its clarity and readability.
> > >
> > > Once more, we appreciate for the time and effort you've dedicated to our paper.

---

### Official Review · Reviewer_eRMs · 2025-07-03

**Rating:** 5
**Confidence:** 4

**Summary:**

The paper introduces MedSG-Bench, the first benchmark that targets sequential medical-image grounding. It aggregates more than nine thousand VQA-style samples from multiple open datasets and multiple modalities and different clinical tasks, organised into eight grounding tasks split between Image-Difference and Image-Consistency paradigms. Baselines show that state-of-the-art general-purpose and medical-specialised MLLMs struggle while the authors’ instruction-tuned MedSeq-Grounder attains sizable gains with training with this dataset.

**Additional Feedback:**

Please address the limitaion and at least provide propritery MLLM performance on the benchmark.

**Dataset Code Accessibility:**

Yes

**Dataset Code Comments:**

The author release both train and bench on huggingface.

**Ethical Comments:**

They are gathering existing open source dataset. Concerns will fall into the source datasets. But it would be nice it author can double check the PHI info are de-identified from the source dataset.

**Ethical Considerations:**

No, there are no or only very minor ethics concerns

**Final Justification:**

I think overall this is a valuable dataset for the community. It reveal the MLLM weakness on small object identification and feature matching. Dataset also shows very clear real-world usage for medical AI.

There are some flaws on writing and experiment setup (basically if 72B model fails we don't expect performance of 7B and lack of proprietary MLLM in the original manuscript). And finetuned model may have some degrees of overfitting due to the in domain nature of train and test.

Overall I would like to recommend accptance.

**Limitations Weaknesses:**

- Question generation: All VQA prompts are paraphrased by GPT-4 only. It might have bias when using single LLM. Suggesting to have some quality check when switch to different llm like claude.
- Baseline comparison missing large portion of proprietary LLM like GPT, claude, Gemini, which should be also compared with table 3. Compare with those model will provide an upper bound and contextualise MedSeq-Grounder’s gains.
- Related work missing: Works on temporal radiology retrieval and lesion progression detection are missing. Example:

**Strengths Contributions:**

- Sequential grounding is crucial for tracking disease progression and cross-modal comparison but lacked a dedicated benchmark; the paper motivates this convincingly with concrete radiology workflows.

- It construct a broad, well-curated dataset making MedSG-Bench much richer than prior single-view efforts. Data standardisation and licence vetting are clearly documented.

-  Eight tasks cover difference detection, multi-view matching, object tracking, etc, reflecting diverse clinical scenarios.

---

> ### Author Rebuttal · Authors · 2025-07-31
>
> We sincerely appreciate the reviewer‘s thoughtful and encouraging comments. We are glad that the motivation behind our study, the carefully curated benchmark, and the comprehensively designed tasks reflecting a wide range of clinical scenarios, have been recognized. Below, we address the reviewer's all questions in detail.
>
> **W1: “Question generation"**
>
> We thank the reviewer for the valuable comment regarding our question generation process. To further investigate whether using a single LLM introduces bias, we have conducted a comprehensive analysis as follows:
>
> **(1) Question Generation and Quality Check:** We generated questions using multiple LLMs, including **GPT-4, Claude, and DeepSeek.** All generated questions were manually reviewed to ensure they preserved the original intent and adhered to the standardized instruction format.
>
> **(2) More Results:** We have additionally evaluated benchmarked models using the questions generated by **GPT-4, Claude, and DeepSeek**, respectively. For your convenience, we also present the average performance (IoU(acc\@0.5)) below, due to space limitations. Detailed per-task results will be provided in the appendix.
>
> **(3) Result Analysis:** The results indicates that, although there are slight performance variations under three LLM-specific prompting settings, these differences remain within an acceptable range and do not affect the key findings presented in the “Main Results” section of the manuscript. We will merge the question data from all LLM-generated sources into a unified evaluation benchmark to mitigate potential bias.
>
> |Model|Size| Avg_gpt |Avg_deepseek| Avg_claude|Avg_ori|
> | ------ |:-:| ------ | ------ | ------ | ------ |
> | Qwen2.5-VL |3B| 10.51 (3.86) | 10.60 (3.85) | 10.31 (9.02) | 10.94 (4.20) |
> | Qwen2.5-VL |7B| 11.25 (15.29) |10.87 (4.13) |11.19 (4.41) | 12.31 (4.90) |
> | Qwen2.5-VL |72B|13.45 (6.37) | 13.35 (6.29)|13.39 (6.41)| 13.35 (6.12) |
> | MiniCPM-V-2_6 |8B|12.72 (4.59)|13.33 (5.13)|12.61 (4.30)|13.24 (5.27)|
> | MiniCPM-O-2_6 |8B| 10.68 (3.85) | 10.34 (3.51) | 10.27 (3.32) |10.12 (3.23)|
> | mPLUG-Owl3 |7B| 10.92 (2.86) | 10.71 (2.69) | 11.04 (2.92) |13.22 (3.19)|
> | Mantis-Idefics2 |8B| 10.33 (4.35) | 10.02 (4.07) | 10.06 (3.91) | 9.90 (3.91)|
> | LLaVA-OneVision |7B| 13.55 (5.51) | 11.59 (3.44) | 12.46 (3.47) |12.39 (3.47)|
> | InternVL2.5 |8B| 7.46 (2.78) | 7.83 (2.72) | 7.13 (2.64) |7.04 (2.56)|
> |Migician|7B|20.31 (11.39)|20.53 (11.91)|20.43 (11.46)|20.29 (11.39)
> |HuatuoGPT-Vision|7B|9.08 (2.71)|9.20 (2.59)|9.18 (2.41)|8.97 (2.36)
> |MedSeq-Grounder|7B|72.68 (79.98)|71.67 (78.76)|72.86 (80.18)|72.55 (79.71)
>
>
> **W2: "Baseline comparison missing large portion of proprietary LLM"**
>
> **(1) Proprietary MLLMs:** We appreciate the reviewer’s comments and have included three leading proprietary MLLMs **(GPT-4o, Claude Sonnet 4, and Gemini 2.5 Pro)** on MedSG-Bench for more comprehensive evaluation. Corresponding results (IoU(acc\@0.5)) are shown below.
>
> **(2) Results Analysis:** Compared to the results reported in **Table 3 of the manuscript**, we found that Gemini-2.5-pro serves as an upper bound. Notably, MedSeq-Grounder still demonstrates substantial performance improvements, underscoring its effectiveness across diverse clinical scenarios.
>
> |Model| RDG| NRDG| MV|OT|VCG| VPG| CMG|RG|Avg|
> | ------ | ------ | ------ | ------ | ------ | ------ | ------ | ------ | ------ |:-:|
> |GPT-4o |2.42 (0.40)|3.45 (0.20)|16.51 (8.62)|28.19 (23.90)|13.18 (4.70)|38.05 (26.40)|16.02 (4.95)|23.08 (18.02)|17.70 (10.60)|
> |Claude Sonnet 4|0.67 (0.00)|0.81 (0.10)|12.56 (3.57)|23.11 (16.50)|6.93 (1.40)|27.44 (13.80)|9.04 (1.80)|19.57 (10.80)|12.51 (5.76)|
> |Gemini 2.5 Pro|9.36 (3.20)|7.29 (2.00)|14.26 (6.71)|19.32 (13.80)|14.94 (10.70)|41.11 (49.20)|24.44 (28.12)|28.12 (22.67)|20.66 (15.61)|
> |MedSeq-Grounder|83.29 (93.20)|83.72 (94.10)|55.03 (60.19)|62.10 (67.20)|74.11 (82.60)|85.25 (98.80)|78.77 (82.75)|60.43 (65.59)|72.55 (79.71)
>
> **W3: "Related work missing"**
>
> We thank the reviewer for pointing out the missing discussion on related work. **While the comment mentioned "Examples:", without listing specific references.** We look forward to further discussion on this topic during the discussion phase of the review process.
>
> In the meantime, we have reviewed recent works related to temporal radiology retrieval and lesion progression detection. These works typically incorporate temporal information either as an explicit feature fused into the model \[1,2,3\] or as a dynamic semantic signal to enhance retrieval \[4\]. Such strategies have shown benefits in downstream tasks such as medical report generation and disease progression analysis. We are preparing to include these representative references to strengthen the related work section in the revised manuscript.
>
> **Ethical Comments: "Double Check the PHI info are de-identified from the source dataset"**
>
> We thank the reviewer for the helpful reminder. We have double-checked all the original datasets used in our work and confirmed that they do not contain any identifiable PHI.
>
> Additionally, we reviewed the corresponding papers and official documentation of each dataset to verify their de-identification process. For example, the MSD \[5\] dataset explicitly states in the “Challenge data sets” section that: "All images were **de-identified** and reformatted to the…", the AMOS22 \[6\] dataset also mentions in the "Data Overview" section that: “Both scanner-generated DICOM and diagnosis reports are collected, **de-identified**, and stored securely.”
>
> **Additional Feedback**
>
> We thank the reviewer once again for your valuable feedback. We hope our detailed responses have addressed the reviewer's concerns, and we will revise the manuscript to enhance the comprehensiveness of MedSG-Bench accordingly.
>
> **Reference**
>
> \[1\] Zhang X, Meng Z, Lever J, et al. Libra: Leveraging temporal images for biomedical radiology analysis[J]. arXiv preprint arXiv:2411.19378, 2024.
>
> \[2\] Mei X, Mao R, Cai X, et al. Medical report generation via multimodal spatio-temporal fusion[C]//Proceedings of the 32nd ACM International Conference on Multimedia. 2024: 4699-4708.
>
> \[3\] Song S, Tang H, Yang H, et al. DDaTR: Dynamic Difference-aware Temporal Residual Network for Longitudinal Radiology Report Generation[J]. arXiv preprint arXiv:2505.03401, 2025.
>
> \[4\] Yang Y, You X, Zhang K, et al. Spatio-Temporal and Retrieval-Augmented Modelling for Chest X-Ray Report Generation[J]. IEEE Transactions on Medical Imaging, 2025.
>
> \[5\] Antonelli M, Reinke A, Bakas S, et al. The medical segmentation decathlon[J]. Nature communications, 2022, 13(1): 4128.
>
> \[6\] Ji Y, Bai H, Ge C, et al. Amos: A large-scale abdominal multi-organ benchmark for versatile medical image segmentation[J]. Advances in neural information processing systems, 2022, 35: 36722-36732.

---

> > ### Comment · Reviewer_eRMs · 2025-08-01
> >
> > Thanks for authors clarification. I truly appreciate the authors amazing work.
> >
> > One more thing to make the contribution clear is that the performance table group by **target-to-image size ratios** as showing in Fig 5. My hypothesis is that this is super helpful for small object identification (<5%).
> >
> > I would also like to have author clarify the limitation of using your model for other task like free text QA. I have test the model and found that model will always output bounding box even I change the text prompt to "Describe the difference of the two images". Finetuned model basically loss its other capability on normal chat/conversation.

---

> > > ### Author Response · Authors · 2025-08-02
> > > **Official Comment by Authors**
> > >
> > > We are delighted to see that the reviewer considers our work an "amazing work", and we sincerely thank the reviewer for the thoughtful and constructive suggestions.
> > >
> > > (1) Small object identification is very important in medical domain. We grouped the targets by their bounding box area ratio into small (0–1%), medium (1–10%), and large (>10%), and evaluated model performance within each group. Per-task results will be included in the revised supplementary. In addition, we also provided the average performance (IoU(acc\@0.5)) below for your convenience.
> > >
> > > |Model|Size| Avg_small |Avg_medium| Avg_large|
> > > | ------ |:-:| ------ | ------ | ------ |
> > > | Qwen2.5-VL |3B|2.27 (1.28)|9.53 (4.66)|24.73 (7.47)|
> > > | Qwen2.5-VL |7B|1.69 (0.48)|8.31 (3.74)|20.13 (7.91)|
> > > |Qwen2.5-VL|32B|3.42 (1.21)|12.01 (5.41)|26.46 (12.78)
> > > | Qwen2.5-VL |72B|3.82 (1.18)|11.96 (5.31)|25.92 (13.25)|
> > > | MiniCPM-V-2_6 |8B|2.93 (0.55)|15.35 (6.40)|24.93 (10.36)|
> > > | MiniCPM-O-2_6 |8B|2.60 (0.38)|10.93 (3.25)|19.82 (7.36)|
> > > | mPLUG-Owl3 |7B|2.41 (0.00)| 14.57 (22.75)|26.86 (8.54)|
> > > | Mantis-Idefics2 |8B|2.92 (1.10)|14.47 (6.72)|12.74 (3.51)|
> > > | LLaVA-OneVision |7B|1.19 (0.00)| 15.33 (4.28)|23.51(7.23)|
> > > |LLaVA-OneVision |72B|3.27 (0.55)|14.68 (3.99)|25.38 (13.87)
> > > |InternVL2|8B|2.24 (1.01)|12.64 (6.09)|18.09 (7.40)
> > > |InternVL2|76B|1.12 (0.32)|7.46 (3.49)|14.39 (8.29)
> > > | InternVL2.5 |8B|2.06 (0.55)|10.24 (4.41)| 9.17 (2.54)|
> > > |InternVL2.5|78B|1.33 (0.38)|8.64 (2.86)|13.85 (5.25)
> > > |InternVL3|8B|1.81 (0.32)|9.06 (2.49)|20.49 (8.50)
> > > |InternVL3|14B|2.23 (0.52)|12.76 (5.12)|19.09 (8.97)
> > > |InternVL3|38B|2.47 (0.61)|11.12 (4.70)|20.71 (9.65)
> > > |InternVL3|78B|1.26 (0.38)|6.80 (2.57)|13.46 (7.11)
> > > |Migician|7B|11.24 (4.72)|22.05 (11.89)|30.69 (20.36)|
> > > |HuatuoGPT-Vision|7B|2.55 (0.35)|11.55 (3.88)|14.20 (2.83)|
> > > |HuatuoGPT-Vision|34B|2.90 (0.43)|11.02 (3.38)|12.93 (2.41)
> > > |MedSeq-Grounder|7B|68.37 (75.83)|69.32 (75.26)|83.88 (92.55)|
> > >
> > > The results indicate that most benchmarked MLLMs struggle with small targets. In contrast, MedSeq-Grounder consistently delivers strong performance across varying target sizes, highlighting its robustness in clinically complex scenarios.
> > >
> > > (2)  We thank the reviewer for pointing this out, and we will clarify this in the revised manuscript. In future work, we plan to continue training the model on broader medical instruction data beyond grounding tasks to enhance its general multimodal capabilities.
> > >
> > > Once again, we truly appreciate the reviewer’s positive and constructive feedback.

---

### Note · Authors · 2025-08-12

Dear AC and reviewers,

We are grateful to the Area Chair for organizing the review process for our paper. We also thank all the reviewers (Reviewer eRMs, ejM7, QurD) for appreciating the motivation behind our study, the carefully curated benchmark, the comprehensively designed tasks reflecting a wide range of clinical scenarios.

During the discussion, we have addressed all questions raised by the reviewers, and the corresponding revisions will be incorporated into the final version of our manuscript. We appreciate the positive feedback and valuable comments that helped to improve our paper. We find that all reviewers have confirmed that they have no further questions or concerns and have a clear understanding of our contributions.

Thank you again for your time and support.

Sincerely,

MedSG-Bench Authors

---

### Decision · Program_Chairs · 2025-09-18

**Decision:**

Accept (spotlight)

**Comment:**

This paper introduces MedSG-Bench, a benchmark for medical image sequence grounding, addressing lesion localization and disease progression tracking across images. The paper defines eight VQA-style tasks across ten modalities, where experiments reveal limitations in existing MLLMs. The paper also builds MedSG-188K for instruction tuning, and develops MedSeq-Grounder to encourage research on understanding medical sequential images.

Strengths:
- Reviewers note that MedSG-Bench is not only the first benchmark dedicated to grounding in sequential medical images that fills a critical gap where existing efforts focus only on single-image scenarios, but also extensive in volume, built from 76 public datasets while spanning 10 imaging modalities.
- Reviewers acknowledged the benchmark's eight well-designed VQA-style tasks, which enables robust assessment of MLLM’s diverse skills (e.g. change detection, cross-modal reasoning) in clinical scenarios.
- Reviewers find it valuable that the paper releases not only MedSG-Bench but also MedSG-188K (a large-scale instruction-tuning dataset) and MedSeq-Grounder, which improves reproducibility as well as benefits the medical AI community.

Weaknesses:

Basically, all weaknesses raised by the reviewers were properly addressed by the authors during the discussion period such as:
- Need to use diverse LLMs to generate VQA prompts to avoid bias
- Need to add more baselines using proprietary MLLMs and medical-domain MLLMs.
- Need to use real pre/post treatment datasets to validate the performance of MedSeq-Grounder
- Need to test clinical windowing instead of min-max normalization of the DICOM images.
- Need for deeper analysis of model failures.

===== FINAL UPDATE FROM DB Track PCs ====

The final decision for this paper has been taken by the program chairs after consultation with the SACs. All Senior Area Chairs have ranked papers according to the feedback from the AC during the review process. We decided to leave the original meta-review to reflect the opinion of the AC in light of the initial discussions with reviewers and SAC.